# ADAPTIVE DESTRUCTION PROCESSES
# FOR DIFFUSION SAMPLERS

## ABSTRACT

This paper explores the challenges and benefits of a trainable destruction process in diffusion samplers – diffusion-based generative models trained to sample an unnormalised density without access to data samples. Contrary to the majority of work that views diffusion samplers as approximations to an underlying continuous-time model, we view diffusion models as discrete-time policies trained to produce samples in very few generation steps. We propose to trade some of the elegance of the underlying theory for flexibility in the definition of the generative and destruction policies. In particular, we decouple the generation and destruction variances, enabling both transition kernels to be learnt as unconstrained Gaussian densities. We show that, when the number of steps is limited, training both generation and destruction processes results in faster convergence and improved sampling quality on various benchmarks. Through a robust ablation study, we investigate the design choices necessary to facilitate stable training. Finally, we show the scalability of our approach through experiments on GAN latent space sampling for conditional image generation.

## 1 INTRODUCTION

Probabilistic inference is concerned with sampling from a distribution defined by an unnormalised density. Contrary to the generative modelling scenario, the learner has access only to an energy function $\mathcal{E}(x) : \mathbb{R}^d \to \mathbb{R}$, *without* access to ground truth samples. The aim is to sample from

$$p_{\text{target}}(x) = e^{-\mathcal{E}(x)}/Z; \quad Z = \int_{\mathbb{R}^d} e^{-\mathcal{E}(x)} \, dx \tag{1}$$

and estimate the (typically intractable) normalising constant $Z$. The classical solution to this problem is given by Monte Carlo methods such as AIS (Neal, 1998) or MCMC methods such as Hamiltonian Monte Carlo (Neal et al., 2011; Hoffman et al., 2014). However, such methods typically require many sampling steps to converge. In addition, some of them rely on access to $\nabla_x \mathcal{E}(x)$, which may not always be available. Some methods use adaptive proposals within Monte Carlo methods, see, *e.g.*, Bugallo et al. (2017); Gabrié et al. (2022); Samsonov et al. (2022); Midgley et al. (2023). This increases the convergence speed of the chain and improves the sampling quality. However, this approach still suffers from slow mixing, especially when the problem dimension $d$ is large and the energy function $\mathcal{E}(x)$ has multiple modes.

In contrast, the *amortised* approach to the sampling problem (1) suggests to train a generative model to approximately sample from $p_{\text{target}}$, offering a more scalable alternative to Monte Carlo or MCMC algorithms. The successes of diffusion models as distribution approximators (Sohl-Dickstein et al., 2015; Ho et al., 2020; Song et al., 2021) motivate the application of diffusion-based techniques to sampling, giving rise to the *diffusion samplers*. Unlike diffusion models, they do not assume access to samples from the target density – only to an energy function that can be queried – and are suitable for cases when the target distribution is represented by a black-box energy function.[1]

There are many algorithms for training diffusion samplers (see §2.1 and related work in Appendix A). One approach, first explored in Zhang & Chen (2022); Vargas et al. (2023), views diffusion samplers as approximators to continuous-time stochastic dynamics. Another approach replaces continuous

---

[1]Diffusion samplers are not to be confused with diffusion models trained from data samples, nor with learnt integrators for pretrained diffusions, which both use quite different algorithms. In particular, methods for training the noising process in diffusion models trained from data (see Appendix A) do not apply to diffusion samplers.

dynamics with a time-discrete Markov chain and trains the diffusion sampler as a discrete policy using reinforcement learning objectives (Lahlou et al., 2023). The assumption of underlying continuous-time dynamics necessitates constraints on the generative and destruction processes: for example, their diffusion coefficients must coincide (see details in Appendix B.1). Discrete-time diffusion samplers have more flexibility: both generation and destruction transition kernels can be flexibly learnt, for example, as Gaussians with mean and variance parametrised by neural networks. In this paper, we take advantage of this flexibility to train both generation and destruction processes, obtaining faster and more accurate samplers. In summary, this paper makes the following contributions:

(1) We consider joint training of generation and destruction processes in diffusion samplers and show that this approach significantly improves sampling quality and normalising constant estimation across various benchmarks. The advantage over fixed destruction increases when the number of sampling steps is small or the energy landscape has narrow modes. To our knowledge, this work is the first to systematically study the effect of a learnable destruction process in diffusion samplers.

(2) This work is the first to implement learnable state-dependent variances of both destruction and generation processes. We show that sampling quality highly benefits from the learnable variance of the generation process, which can be decoupled from that of the destruction process and optimized simultaneously – a unique feature of the discrete-time formulation of diffusion samplers.

(3) We point out practical guidelines that are crucial for stable and effective joint training of generation and destruction processes – such as shared backbones, separate optimisers, target networks, and prioritised replay buffers – and validate them through a comprehensive ablation study.

(4) Finally, we show the scalability of our approach to high-dimensional problems through experiments on text-conditional sampling in the latent space of a pretrained StyleGAN3 (Karras et al., 2021), demonstrating quantitative and qualitative benefits of training the destruction process.

## 2 BACKGROUND

### 2.1 DIFFUSION SAMPLERS IN CONTINUOUS TIME

We consider the sampling problem (1) in a setting when the learner has an access only to the energy function $\mathcal{E}(x)$.[2] We want to sample from $p_{\text{target}}$ by simulating the dynamics of a process characterised by a (forward or generative) Itô SDE:

$$\mathrm{d}\overrightarrow{X_t} = \overrightarrow{f_\theta}(\overrightarrow{X_t}, t)\, \mathrm{d}t + g(t)\, \mathrm{d}\overrightarrow{W_t}, \quad \overrightarrow{X_0} \sim p_0, \tag{2}$$

where $p_0$ is a tractable distribution (*e.g.*, Gaussian or Dirac). The drift function $\overrightarrow{f_\theta}$ is a neural network with parameters $\theta$, and we want to find $\theta$ such that $\overrightarrow{X_1} \sim p_{\text{target}}$, *i.e.*, the terminal samples $\overrightarrow{X_1}$ are distributed as $p_{\text{target}}$. There are many SDEs of the form in (2) stochastically transporting $p_0$ to $p_{\text{target}}$. To set a learning target for $\theta$, we consider a backward (or destructive) SDE that is constructed so as to terminate in $p_0$ when initialised at any $p_{\text{target}}$:

$$\mathrm{d}\overleftarrow{X_t} = \overleftarrow{f_\varphi}(\overleftarrow{X_t}, t)\, \mathrm{d}t + g(t)\, \mathrm{d}\overleftarrow{W_t}, \quad \overleftarrow{X_1} \sim p_{\text{target}}. \tag{3}$$

Given a fixed process (3), there is a unique forward SDE of the form (2) that defines the same process, *i.e.*, the processes $\overrightarrow{X_t}$ and $\overleftarrow{X_t}$ coincide. In particular, the marginal distributions at time 1 would then coincide, so $\overrightarrow{X_1} \sim p_{\text{target}}$. Thus, training objectives aim to enforce *time reversal* – to minimise some divergence between the generation and destruction processes with respect to their parameters.

One approach to solving this problem, related to continuous-time stochastic control, involves writing a divergence functional defined in continuous time. This functional may compute a discrepancy between path space measures (*i.e.*, distributions over trajectories, see Nüsken & Richter (2021); Vargas et al. (2022); Zhang & Chen (2022); Vargas et al. (2024); Berner et al. (2024)), which can then be approximately minimised in a time discretisation. Alternatively, one can use time-local objectives involving the processes' marginal distributions at intermediate times $t$, derived from the Fokker-Planck-Kolmogorov or Hamilton-Jacobi-Bellman PDEs (see Nüsken & Richter (2023); Máté & Fleuret (2023); Sun et al. (2024)).

Another approach is to immediately replace the generation and destruction SDEs by corresponding discrete-time Markov chains and to learn the dynamics given a fixed number of steps. The discretisa-

---

[2]For absolutely continuous densities, we abuse notation and use the distribution and density interchangeably. Further, all claims about SDEs hold under basic regularity conditions, *e.g.*, those in Berner et al. (2025), §B.1.

tion may be defined by Euler-Maruyama integration of the SDEs (see §2.2). One can then borrow objectives from reinforcement learning to enforce time reversal of discrete-time processes. This approach was notably explored in Lahlou et al. (2023); Sendera et al. (2024) and is in theory more general, as it allows more flexibility in the choice of transition kernels (Phillips & Cipcigan, 2024; Volokhova et al., 2024).

The two approaches become equivalent in the continuous-time limit: as the time discretisation becomes finer, objectives for training discrete-time transition kernels approach their continuous-time counterparts, justifying the use of varying time discretisations for training (Berner et al., 2025). However, learning in continuous time presents a notable limitation: it necessitates that variances of both generation and destruction processes remain the same. Otherwise, divergence functionals cannot be defined, as we demonstrate in Appendix B.1. In §3.1, we will study how the assumption of fixed variances can be relaxed in the discrete-time approach and how this can be beneficial for modelling.

## 2.2 DIFFUSION SAMPLERS IN DISCRETE TIME

In order to learn the time-discrete variant of (2) we use a $T$-step Euler-Maruyama scheme to obtain a Markov chain: $X_0 \rightarrow X_{\Delta t} \rightarrow \ldots \rightarrow X_1$; $\Delta t = 1/T$.[3] The discrete dynamics can be written as:

$$X_{t+\Delta t} = X_t + f_\theta(X_t, t)\,\Delta t + g(t)\sqrt{\Delta t}\,\xi_t, \quad X_0 \sim p_0(X_0),\ \xi_t \sim \mathcal{N}(0, I_d) \qquad (4)$$

with the conditional probability density for each timestep being

$$\overrightarrow{p_\theta}(X_{t+\Delta t} \mid X_t) = \mathcal{N}(X_{t+\Delta t}; X_t + f_\theta(X_t, t)\,\Delta t,\ g(t)^2 \Delta t\, I_d)\,. \qquad (5)$$

The forward and reverse transition kernels $\overrightarrow{p_\theta}$ and $\overleftarrow{p_\varphi}$ induce joint distributions over $\mathbf{X} = (X_0, X_{\Delta t}, \ldots, X_1)$, corresponding to time-discrete variants of (2) and (3)[4]:

$$\overrightarrow{p_\theta}(\mathbf{X}) = p_0(X_0)\overrightarrow{p_\theta}(X_{\Delta t, \ldots, 1} \mid X_0) = p_0(X_0) \prod_{i=1}^{T} \overrightarrow{p_\theta}(X_{i\Delta t} \mid X_{(i-1)\Delta t}) \qquad (6)$$

$$\overleftarrow{p_\varphi}(\mathbf{X}) = p_{\text{target}}(X_1)\overleftarrow{p_\varphi}(X_{0, \ldots, (T-1)\Delta t} \mid X_1) = p_{\text{target}}(X_1) \prod_{i=1}^{T} \overleftarrow{p_\varphi}(X_{(i-1)\Delta t} \mid X_{i\Delta t}), \qquad (7)$$

where if $p_0$ is Dirac the transition $\overleftarrow{p_\varphi}(X_0 \mid X_{\Delta t})$ is understood to be Dirac as well (and both are taken to have density 1 in the density ratios representing Radon-Nikodym derivatives in §3.2). When both generation and destruction kernels are trained we expect the following equality to hold:

$$p_0(X_0)\overrightarrow{p_\theta}(X_{\Delta t, \ldots, 1} \mid X_0) = p_{\text{target}}(X_1)\overleftarrow{p_\varphi}(X_{0, \ldots, (T-1)\Delta t} \mid X_1) \qquad (8)$$

or $p_0\overrightarrow{p_\theta} = p_{\text{target}}\overleftarrow{p_\varphi}$ for brevity.

The vast majority of past work focuses on learning the generation process given a fixed destruction process (see Appendix A). We propose to exploit the flexibility of the time-discrete formulation of diffusion samplers to parametrise both drift and variance of generation and destruction processes using neural networks. We further detail our approach in §3.2 and show experimental findings in §4. There are previous works that consider learnable drifts of destruction processes (Richter & Berner, 2024; Blessing et al., 2025b). However, Richter & Berner (2024) observe that such design choice does not provide consistent improvements and can lead to numerical instabilities in the training procedure. In our experiments (§4.2), we consider an approach with trainable drifts and fixed variances as a baseline, showing that with the use of our parametrisation and several stabilisation techniques (§3.3) it can offer improvements over training with a fixed destruction process. While such an approach does offer small improvements, we also show that the main benefits of learnable destruction processes are observed when *both* drifts and variances are learnt.

## 3 METHODOLOGY

### 3.1 RELAXING CONSTRAINTS ON TRANSITIONS

We first propose to relax the constraint that the generation and destruction processes arise from a pair of SDEs (2) and (3) with fixed time-dependent variance. We begin with the destruction process that is the reversal of time-discretised Brownian motion with fixed diffusion coefficient $\sigma$ starting from $p_0 = \delta_0$, as considered by Zhang & Chen (2022) and many later works. For this process, the

---

[3]For notational clarity we assume a uniform discretisation, but all derivations hold for variable time steps.

[4]The reverse-time discretisation can use the reverse Euler-Maruyama scheme, or (as in Zhang & Chen (2022) and other works, including ours) the reversal of a forward-time Euler-Maruyama scheme, which may not coincide – in particular, this yields the multiplier $\frac{t}{t+\Delta t}$ on the variance in (10). The two coincide as $\Delta t \rightarrow 0$; see, *e.g.*, Berner et al. (2025, Prop. 3.5).

transitions have the form:

$$\overrightarrow{p_\theta}(X_{t+\Delta t} \mid X_t) = \mathcal{N}(X_{t+\Delta t}; X_t + f_\theta(X_t, t)\,\Delta t,\ \sigma^2 \Delta t\, I_d), \tag{9}$$

$$\overleftarrow{p}(X_t \mid X_{t+\Delta t}) = \mathcal{N}\left(X_t; \frac{t}{t + \Delta t}X_{t+\Delta t},\ \frac{t}{t + \Delta t}\sigma^2 \Delta t\, I_d\right), \tag{10}$$

where $\overleftarrow{p}(x_0 \mid x_{\Delta t})$ is understood as Dirac $\delta_0(x_0)$.

We relax this constraint and allow both generation and destruction processes to be Gaussian with arbitrary means and variances, while maintaining the requirement that the destruction process leads to $p_0$ at the last step. This is done by learning the generation and destruction means and variances as corrections to those in (9) and (10), respectively. By introducing corrections we get the following generative transition kernel:

$$\overrightarrow{p_\theta}(X_{t+\Delta t} \mid X_t) = \mathcal{N}(X_{t+\Delta t}; X_t + f_\theta(X_t, t)\,\Delta t,\ \mathrm{diag}(\gamma_\theta(X_t, t))\sigma^2 \Delta t\, I_d),$$
$$\gamma_\theta(X_t, t) = \exp\left\{C_1 \tanh\left(\mathrm{NN}_\theta^{(1)}(X_t, t)\right)\right\} \tag{11}$$

and the following destruction kernel:

$$\overleftarrow{p_\varphi}(X_{t-\Delta t} \mid X_t) = \mathcal{N}\left(X_{t-\Delta t}; \mathrm{diag}(\alpha_\varphi(X_t, t))\frac{t - \Delta t}{t}X_t,\ \mathrm{diag}(\beta_\varphi(X_t, t))\frac{t - \Delta t}{t}\sigma^2 \Delta t\, I_d\right),$$
$$\alpha_\varphi(X_t, t) = 1 + C_2 \tanh\left(\mathrm{NN}_\varphi^{(2)}(X_t, t)\right),\quad \beta_\varphi(X_t, t) = 1 + C_2 \tanh\left(\mathrm{NN}_\varphi^{(3)}(X_t, t)\right), \tag{12}$$

where the $\mathrm{NN}^{(i)}$ represent neural networks with $d$-dimensional outputs and $C_1, C_2$ are constants. Such a parametrisation guarantees that the generation process variance remains between $\frac{1}{C_1}$ and $C_1$ times that of the baseline one in (9), while the multiplicative deviations of the destruction process mean and variance from the basic ones in (10) are bounded between $1 - C_2$ and $1 + C_2$. We empirically found such a combination to work better than using the same parametrisation for both generation and destruction processes. This parametrisation does not impose any other constraints on either process – they are no longer discretisations of SDEs defining absolutely continuous path space measures.

In theory, the transitions could be more general than Gaussian, as long as their density is tractable to sample and differentiate. The flexibility of this approach allows easy generalisation to mixtures of Gaussians, transition kernels on manifolds, or even non-parametric distributions.

**Why train the variance of the generation process?** When the number of time discretisation steps is large, the reverse of a fixed destruction process $\overleftarrow{p_\varphi}$ is well approximated by Gaussian transitions with the same variances. However, when the number of steps is small, the discrete-time generation process should be able to adapt to the discretisation of the destruction process. A preliminary experiment in Sendera et al. (2024) showed that allowing for the variance of the generation process to be learnt when the destruction process is fixed can improve sampling quality. Thus, we consider the combination of a fixed destruction process and generation process with learnable mean and variance as one of the baselines in our experiments.

**Why train the destruction process?** The ability to train the destruction process increases the degrees of freedom in the feasible space of solutions $(p_0\overrightarrow{p_\theta}, p_{\text{target}}\overleftarrow{p_\varphi})$ when minimising the discrepancy in the time-reversal condition (8). Modulo optimisation errors, the solution with a trainable destruction process is guaranteed to be at least as good as the one with a fixed destruction process: the latter is a special case of the former, where the learnt correction to the destruction process is zero.

### 3.2 Training objectives for generation and destruction processes

Learning discrete Markov process requires optimising a divergence $\mathbb{D}(p_0\overrightarrow{p_\theta}\|p_{\text{target}}\overleftarrow{p_\varphi})$ between two distributions over $\mathbf{X} = (X_0, X_{\Delta t}, \ldots, X_1)$ that enforces equality in (8). One option is the reverse KL:

$$\mathbb{D}_{\mathrm{KL}}(p_0\overrightarrow{p_\theta}\|p_{\text{target}}\overleftarrow{p_\varphi}) = \mathbb{E}_{\mathbf{X}\sim\overrightarrow{p_\theta}(\mathbf{X})}\left[\log \frac{p_0(X_0)\overrightarrow{p_\theta}(X_{\Delta t,\ldots,1} \mid X_0)}{\exp(-\mathcal{E}(X_1))\overleftarrow{p_\varphi}(X_{0,\ldots,(T-1)\Delta t} \mid X_1)}\right] + \log Z. \tag{13}$$

Here the log-normalising constant $\log Z$ does not affect the optimisation. Methods that propose to optimise this objective include Zhang & Chen (2022); Vargas et al. (2023). They use the reparametrisation trick to rewrite (13) as an expectation over the noises used in integration (the $\xi_t$ in (4)). Although this scheme yields an unbiased estimator, it requires backpropagating though the simulation

of the generation process. Alternatively, one can use second-moment divergences of the form

$$\mathbb{D}_{\tilde{p}}(p_0\overrightarrow{p_\theta}\|p_{\text{target}}\overleftarrow{p_\varphi}) = \mathbb{E}_{\mathbf{X}\sim\tilde{p}(\mathbf{X})}\left[\log\frac{p_0(X_0)\overrightarrow{p_\theta}(X_{\Delta t,\dots,1}\mid X_0)}{\exp(-\mathcal{E}(X_1))\overleftarrow{p_\varphi}(X_{0,\dots,(T-1)\Delta t}\mid X_1)}+\log\hat{Z}\right]^2 \quad (14)$$

where $\tilde{p}$ is some full-support proposal distribution not necessarily equal to $\overrightarrow{p_\theta}$ and $\log\hat{Z}$ is a scalar. This scalar absorbs the (unknown) true normalising constant $Z$ of $p_{\text{target}}$, allowing to use the unnormalised log-density $-\mathcal{E}(X_1)$ in place of $\log p_{\text{target}}(X_1) = -\mathcal{E}(X_1) - \log Z$ in the denominator. The scalar $\log\hat{Z}$ can be a learnt parameter (yielding the loss called **trajectory balance** (**TB**; Malkin et al., 2022)) or analytically computed to minimise the loss on each batch of training trajectories (yielding the log-variance or **VarGrad** loss (Richter et al., 2020)). In either case, when the loss is minimised to 0, $\log\hat{Z}$ becomes equal to the true log-partition function $\log Z$.

Unlike the KL divergence (13), which is an expectation over a distribution depending on $\theta$, second-moment divergences avoid backpropagating through sampling, allowing the use of off-policy exploration techniques to construct the proposal distribution $\tilde{p}$. On the other hand, when $\tilde{p} = p_\theta$, the expected gradient of $\mathbb{D}_{\tilde{p}}$ with respect to $\theta$ – *but not with respect to $\varphi$* – remarkably coincides with that of $\mathbb{D}_{\text{KL}}$ (Malkin et al., 2023).

The divergence in (13) can be used for optimisation of $\overleftarrow{p_\varphi}$ in addition to $\overrightarrow{p_\theta}$. Notice that this is equivalent to log-likelihood maximisation with respect to $\varphi$, since the gradient of (13) takes the form:

$$\nabla_\varphi\mathbb{D}_{\text{KL}}(p_0\overrightarrow{p_\theta}\|p_{\text{target}}\overleftarrow{p_\varphi}) = \nabla_\varphi\mathbb{E}_{\mathbf{X}\sim\overrightarrow{p_\theta}(\mathbf{X})}\left[-\log\left(\overleftarrow{p_\varphi}(X_{0,\dots,(T-1)\Delta t}\mid X_1)\right)\right] \quad (15)$$

$$= \nabla_\varphi\mathbb{E}_{\mathbf{X}\sim\overrightarrow{p_\theta}(\mathbf{X})}\left[\sum_{i=1}^{T}-\log\overleftarrow{p_\varphi}(X_{(i-1)\Delta t}\mid X_{i\Delta t})\right] \quad (16)$$

In the more general GFlowNet setting, such approach for learning the destruction process is called **trajectory likelihood maximisation** (**TLM**; Gritsaev et al., 2025).

Table 1 summarises the possible combinations of objectives for destruction and generation policies. In §4 we numerically study the behaviour of the aforementioned objectives.

Table 1: Summary of the objectives studied for learning the generation and destruction processes.

| $\overleftarrow{p_\varphi}$ (destruction) ↓ | $\overrightarrow{p_\theta}$ (generation) | |
| --- | --- | --- |
| | 2nd moment | Reverse KL |
| *Fixed* | Richter & Berner (2024), Sendera et al. (2024) | PIS (Zhang & Chen, 2022), DDS (Vargas et al., 2023) |
| 2nd moment | $\text{TB}_{\theta,\varphi}$ | $\text{PIS}_\theta + \text{VarGrad}_\varphi$ |
| Rev. KL | $\text{TB}_\theta + \text{TLM}_\varphi$ | $\text{PIS}_\theta + \text{TLM}_\varphi$ |

### 3.3 TECHNIQUES

FOR STABLE JOINT OPTIMISATION

We found that simultaneously training generation and destruction processes requires careful design. In this section we review the utilised techniques that facilitate stable and effective training. Detailed descriptions are presented in Appendix C, and a numerical study is carried out in §4.3.

**Shared backbone.** It is possible to parametrise generation and destruction processes with separate neural networks with linear heads for mean and log-variance or to use a shared backbone with different heads for each process. We motivate the latter parametrisation by the idea that sharing representations between the two processes facilitates faster convergence. This design choice was also explored and motivated in GFlowNet literature (Malkin et al., 2022; Gritsaev et al., 2025).

**Separate optimisers.** When representations are shared, the backbone parameters receive gradient signals from both the losses used for training $\theta$ and $\varphi$. Their gradients may have different scales, especially when different training objectives are employed for them, which may cause instability if both gradient updates use the same adaptive-momentum optimiser. Thus, we use separate optimisers for the generation and destruction process losses.

**Learning rates.** Since the destruction process sets a learning target for the generation process, making it learnable causes this target to become nonstationary, leading to instabilities in optimisation (although the joint optimisation has no saddle point at the global optimum). Thus, tuning relative learning rates is critical for stable training. The same effect was studied in Gritsaev et al. (2025) for learnable destruction processes in GFlowNets.

**Target network.** Target networks are a well-known RL stabilisation technique (Mnih et al., 2015) that replaces regression targets in a loss function by variants smoothed over training time (by a

lagging update, *e.g.*, using exponential moving average (Silver et al., 2014)). We use a target network for $\varphi$ when computing the loss gradient for $\theta$ and vice versa.

**Prioritised experience replay.** Replay buffers are another common RL technique that allows to prevent forgetting by resampling previously used trajectories. We use prioritised experience replay (PER; Schaul et al., 2016) with prioritisation by the loss, which was also utilised for GFlowNet training by Tiapkin et al. (2024). The replay ratio is a hyperparameter that determines the number of gradient updates per one newly sampled trajectory.

**Better exploration in off-policy methods.** Training with second-moment divergences allows a flexible choice of behaviour policy $\tilde{p}$ (see (14)). We use the existing techniques proposed by Lahlou et al. (2023); Sendera et al. (2024) to facilitate exploration during training, namely, increasing the variance of the generation process for sampling trajectories for training, as well as local search based on Langevin dynamics.

### 3.4 CONNECTION TO REINFORCEMENT LEARNING

In this section we briefly discuss a fact that provides important motivation and context for some of the techniques mentioned in §3.3: it is possible to formulate the training of a diffusion sampler in discrete time as an RL problem with entropy regularisation (also called soft RL (Neu et al., 2017; Geist et al., 2019)). Although, up to our knowledge, it was never directly stated in this form in previous literature on diffusion samplers, this fact is well-understood, and a number of works on diffusion samplers frame their training as a stochastic control task in continuous time (Zhang & Chen, 2022; Vargas et al., 2024; Berner et al., 2024). The equivalence with soft RL in discrete time can be directly shown by combining the analysis of Tiapkin et al. (2024); Deleu et al. (2024), which show that GFlowNet training is equivalent to a soft RL task, and Lahlou et al. (2023); Sendera et al. (2024); Berner et al. (2025), which show that discrete-time diffusion samplers are a special case of continuous GFlowNets. We provide the details and the proof in Appendix B.2. This connection gives an important motivation for using such techniques as target networks (Mnih et al., 2015), replay buffers (Schaul et al., 2016), and various exploration methods, which are all well-studied in the RL literature, for the training of diffusion samplers.

## 4 EXPERIMENTS

### 4.1 SETUP ON STANDARD BENCHMARKS

**Energies.** We consider several target densities which are commonly used in the diffusion samplers literature, as well as more challenging variants of these distributions (see Appendix D.1 for details):

- **Gaussian mixture models (GMM).** In addition to the simple mixture of 25 regularly spaced Gaussians (**25GMM**) and 40 irregularly spaced Gaussians (**40GMM**) in $\mathbb{R}^2$, we also consider a more difficult $\mathbb{R}^3$ variant (**125GMM**) and a mixture of 25 Gaussians with anisotropic covariance matrices ((**Slightly) Distorted 25GMM**).
- **Funnel** (Neal, 2003). We consider two 10-dimensional funnel distributions from past work. Two versions (termed **Easy/Hard**) have been considered in the literature due to an error in the definition originating in Zhang & Chen (2022) (see (Sendera et al., 2024, §B.1)); we study both.
- **Manywell** (Noé et al., 2019). This is a distribution on $\mathbb{R}^{32}$ with $2^{16}$ modes that are separated by energy barriers, making it challenging for diffusion samplers to avoid mode collapse. Just as for GMMs, we also define a **Distorted** variant that does not factor into 16 independent 2D distributions.
- **VAE** (Sendera et al., 2024). The task is to sample from the 20-dimensional latent posterior distribution, $p(z|x)$, of a VAE trained on MNIST with a target density $p(z|x) \propto p(x|z)p(z)$.

**Metrics.** Following the literature, we primarily use ELBO and EUBO metrics to evaluate the quality of trained samplers (Blessing et al., 2024). The two metrics lower- and upper-bound the log-partition function of the target distribution, respectively, and EUBO requires access to ground truth samples, which can be obtained exactly for GMMs and Funnel and approximately for Manywell. (See Appendix D.2 for definitions.) We also compute the 2-Wasserstein distance between generated and ground truth samples as a more direct measure of sample quality.

**Algorithms compared.** We use two objectives for optimisation of the generation process: trajectory balance (TB$_\theta$, (Malkin et al., 2022)) and reverse KL (equivalent to PIS (Zhang & Chen, 2022) and so denoted PIS$_\theta$). We compare four settings with each generation process objective:

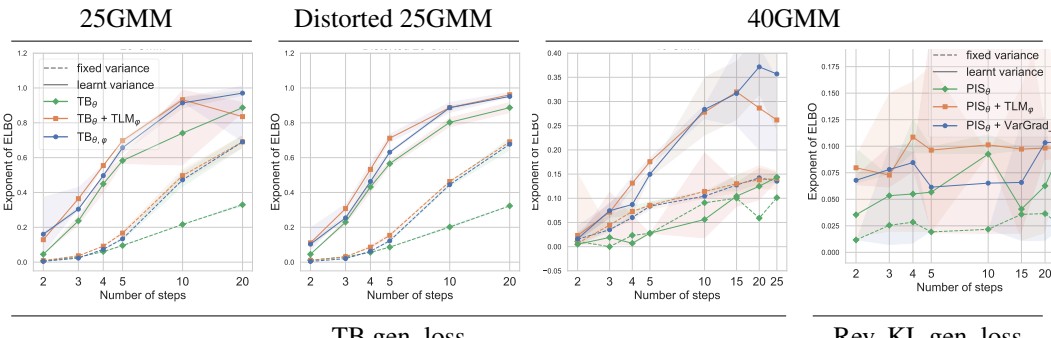

Figure 1: We compare performance of $TB_\theta$, $TB_\theta + TLM_\varphi$, $TB_{\theta,\varphi}$ with both fixed and learnable variances on three GMM targets. The rightmost plot shows PIS-like generation process objectives on **40GMM**. Results are compared using ELBO. Mean and std over 3 seeds, collapsed runs excluded.

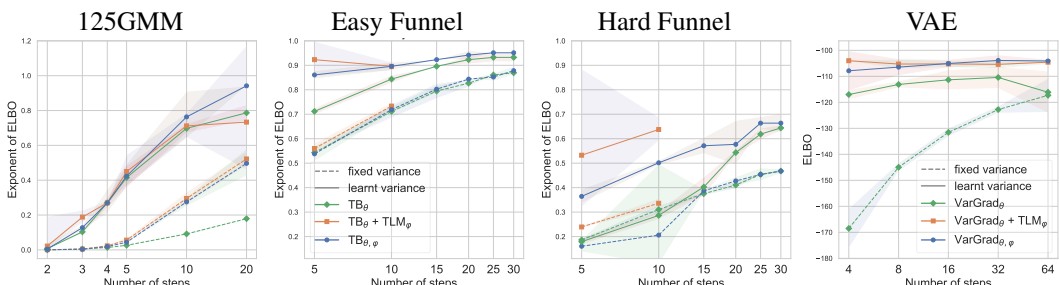

Figure 2: The same plots as Fig. 1 for **125GMM**, **Easy Funnel**, **Hard Funnel**, and **VAE**.

- Fixed generation variances, fixed destruction process (as in most prior work);
- Learnable generation variances, fixed destruction process;
- Learnable generation variances, destruction process learnt using reverse KL ($TLM_\varphi$) or TB ($TB_\varphi$). (When the generation process is learnt using reverse KL, we replace $TB_\varphi$ by $VarGrad_\varphi$, since the log-normalising constant is not learnt.)
- Learnable means for both generation and destruction processes, but both variances fixed (also with both TB and TLM objectives for the destruction process).

**Training.** We train all samplers with a varying number of time discretisation steps $T$. The same number of steps is used for training and for sampling. We use uniform time discretisation for all environments except mixtures of Gaussians, which use a harmonic discretisation scheme (Berner et al., 2025). Notably, unlike some past work, we do *not* use the physics-informed Langevin parametrisation from Zhang & Chen (2022) in the generation process drift, so as to reduce training cost and study the effects of our modifications in isolation. All details can be found in Appendix D.3.

### 4.2 RESULTS ON STANDARD BENCHMARKS

Representative results are shown in Figs. 1 and 2 and full tables are presented in Appendix F. Losses in the legends including only $\theta$ correspond to the setting of a fixed destruction process, while losses including $\theta$ and $\varphi$ correspond to using learnable destruction processes (both with either fixed or learnt variances).

**Learnt variances improve sampling.** Learnable variance of the generation process significantly improves the performance of the sampler, especially when the number of generation steps is small. As shown in Fig. 1, models with learnable generation variance dramatically outperform ones with fixed variance: on some energies, learnt-variance samplers with as few as 5 generation steps outperform 20-step samplers with fixed variance. Second, the importance of trainable variance is clearly observable in complex environments or environments with high distortions (Fig. 2). If the variance is fixed, the sampler is likely to struggle with correctly sampling narrow modes: in particular, the addition of noise of variance $\sigma^2 \Delta t$ on the last step imposes a smoothness constraint on the modelled distribution. Conversely, learning the magnitude of added noise allows to capture the shape of such modes.

**Off-policy losses are superior to differentiable simulation.** The off-policy TB loss with exploratory behaviour policy is on par with or better than the PIS loss for learning the generation

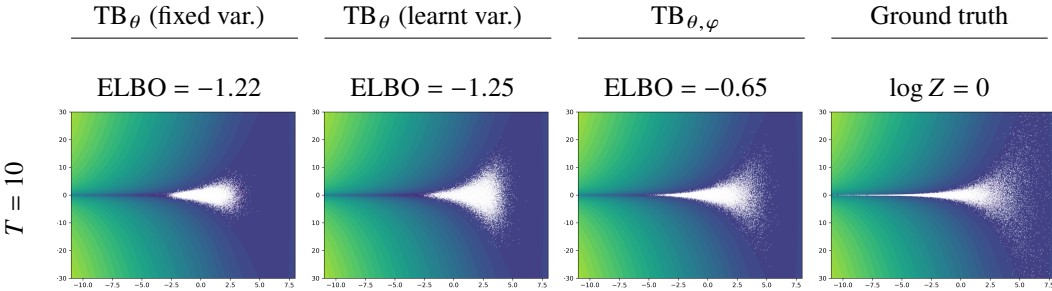

Figure 3: First two dimensions of target energy and samples from diffusion samplers trained on the Hard Funnel energy. Samplers with fixed destruction process, and especially those with fixed generation process variance, struggle to fit the narrow tails accurately.

process in all cases, as shown in the right two panels of Fig. 1 and Appendix F. This is consistent with findings in past work, such as Richter & Berner (2024); Sendera et al. (2024); Kim et al. (2025b). Extending these findings, find that this improvement is maintained when generation variances and the destruction process are learnt. Moreover, the TB loss is more memory-efficient than PIS, as it does not require storing the entire computation graph of the sampling trajectory for backpropagation.

**Learning the destruction process is beneficial.**    Learning the destruction process yields an improvement over models with fixed destruction process and learnt generation variance on all tasks, although the improvement is often less pronounced when the number of sampling steps is large (*e.g.*, on the Funnel densities in Fig. 2), presumably because the reverse of a fixed destruction process is better modelled by Gaussian transitions when the number of steps is large. See Fig. 3, showing that learnt variance helps to model the narrow part of the funnel.

**TB is preferable to TLM for learning destruction.**    Extending the results of Gritsaev et al. (2025) in discrete cases, we find that training the destruction process with the TLM loss is typically superior to TB when the number of steps is small. However, the TLM loss is unstable and often leads to divergent training when the number of steps is large (see Appendix F). We note that, even at optimality, the TLM loss gradient has nonzero variance, while the TB loss gradient is zero for all trajectories at the global optimum, which may explain TB's greater stability.

**Learning only means of generation and destruction processes performs subpar.**    Learning the destruction process mean when using fixed variances for both processes, as done in diffusion bridge samplers in Richter & Berner (2024), offers a subtle benefit at few steps that grows as the number of steps increases (Fig. 1). However, this advantage disappears on Funnel, where learning the variances is critical (Fig. 2) for accurate modelling the target distribution. Ultimately, the best results are obtained in all cases when both processes have learnable means and variances.

**Findings generalise to conditional tasks.**    We extend our findings to the problem of conditional sampling from the latent VAE posterior studied in Sendera et al. (2024). Our experiments confirm that the learnable variance and destruction process greatly improve results, especially when the number of steps is low (Fig. 2). We present results for VarGrad (Richter et al., 2020) on this task, as this method performs well on conditional generation problems where learning the normalization constant for each condition for TB is difficult (see Sendera et al. (2024) for a discussion).

### 4.3 ABLATION STUDY

In this subsection we discuss the results of the ablation study to test the design choices outlined in §3.3. The numerical results are presented in Table 2.

We find that using a shared backbone drastically increases the quality of the sampler, but it is optimal to use separate optimisers for the two processes even when the backbone is shared. We empirically find that the performance of samplers is sensitive to learning rates, and thus they must be carefully tuned for each environment. We use equal learning rates for $\theta$ and $\varphi$ for GMM distributions. However, for more complex environments the destruction policy learning rate must be smaller than that of the generation policy. For instance, in Hard Funnel, a $10^3$ times smaller learning rate for $\varphi$ is optimal, and for Manywell, the optimal ratio is $10^4$ or $10^5$. The best replay ratio in our setup is 3, and we use this value for all our experiments. Moreover, using target networks increases the stability of the training and the final quality of the sampler.

Table 2: Different configurations compared by ELBO, EUBO and 2-Wasserstein between generated and ground truth samples on 40GMM. Mean and std over 5 runs are specified.

| Ablation | Setting | ELBO ($\uparrow$) | EUBO ($\downarrow$) | 2-Wasserstein ($\downarrow$) |
|---|---|---|---|---|
| *(Optimal configuration)* | $TB_{\theta,\varphi}$, shared backbone, separate optimisers, $lr_\varphi = lr_\theta$, replay ratio = 3, target network | $-1.89_{\pm0.10}$ | $5.06_{\pm0.35}$ | $18.71_{\pm0.63}$ |
| Fixed variance | $TB_{\theta,\varphi}$, fixed generation and destruction var. | $-2.49_{\pm0.05}$ | $101.84_{\pm11.23}$ | $30.15_{\pm2.30}$ |
| Learning only generation | $TB_\theta$, fixed generation var. | $-2.47_{\pm0.06}$ | $105.1_{\pm27.11}$ | $32.56_{\pm2.3}$ |
| | $TB_\theta$, learnt generation var. | $-3.5_{\pm0.11}$ | $37.02_{\pm45.42}$ | $20.37_{\pm6.14}$ |
| Models and optimisers | Separate backbones | $-6.03_{\pm1.15}$ | $26.71_{\pm2.70}$ | $28.29_{\pm0.49}$ |
| | Single optimizer | $-2.08_{\pm0.15}$ | $13.05_{\pm4.95}$ | $20.40_{\pm0.71}$ |
| Learning rates | $lr_\varphi = 0.10 \times lr_\theta$ | $-2.06_{\pm0.20}$ | $6.55_{\pm3.48}$ | $16.79_{\pm1.71}$ |
| | $lr_\varphi = 0.01 \times lr_\theta$ | $-2.69_{\pm0.94}$ | $61.43_{\pm30.20}$ | $21.77_{\pm3.68}$ |
| Replay ratio | Replay ratio = 1 | $-1.96_{\pm0.13}$ | $18.59_{\pm11.72}$ | $20.86_{\pm1.10}$ |
| | Replay ratio = 2 | $-2.28_{\pm1.05}$ | $23.53_{\pm15.92}$ | $16.74_{\pm1.10}$ |
| | Replay ratio = 5 | $-1.91_{\pm0.18}$ | $6.08_{\pm1.46}$ | $16.74_{\pm1.10}$ |
| | Replay ratio = 9 | $-3.43_{\pm1.59}$ | $21.24_{\pm21.20}$ | $16.79_{\pm1.71}$ |
| Target network | No target network | $-2.07_{\pm0.09}$ | $10.36_{\pm2.02}$ | $20.13_{\pm1.10}$ |

## 4.4 SCALABILITY: GAN LATENT SPACE SAMPLING FOR CONDITIONAL IMAGE GENERATION

To validate our main findings in a high-dimensional setting, we consider the setup proposed in Venkatraman et al. (2025). Let $g_\psi : \mathbb{R}^{d_{\text{latent}}} \to \mathbb{R}^{d_{\text{data}}}$ be a pretrained GAN (Goodfellow et al., 2014) generator, and $r(x, y)$ be some positive-valued function operating on data points $x \in \mathbb{R}^{d_{\text{data}}}$ and conditions $y$. The task is to sample latent vectors $z$ from the distribution defined by the energy $\mathcal{E}(z) = -\log p_{\text{prior}}(z) - \beta \log r(g_\psi(z), y)$, where $p_{\text{prior}}$ is $\mathcal{N}(0, I)$. The decoded samples $g_\psi(z)$ then follow the posterior distribution, proportional to the product of the GAN image prior and the tempered constraint $r(x, y)^\beta$. In our experiment we use StyleGAN3 (Karras et al., 2021) trained on the $256 \times 256$ FFHQ-256 dataset (Karras et al., 2019) with $d_{\text{latent}} = 512$. We take $y$ to be a text prompt and $\log r(x, y)$ to be the ImageReward score (Xu et al., 2023); thus our aim is to sample latents that produce faces aligned with the specified text prompt. Note that the GAN itself is unconditional, and text-conditioning is only achieved via sampling from the distribution defined through ImageReward, where each prompt defines a distinct energy function. An optimal model should trade off between high reward and high diversity.

We train diffusion samplers with only $T = 5$ sampling steps to sample from the specified target distribution in $\mathbb{R}^{d_{\text{latent}}}$, comparing the approach from Venkatraman et al. (2025) ($TB_\theta$ with fixed generation variances) to $TB_{\theta,\varphi}$ where both processes have learnable means and variances. In addition to ELBO, we report average ImageReward score and diversity, measured as average cosine distance of CLIP (Radford et al., 2021) embeddings for 128 generated images. We train separate models across 7 different prompts and find improvements in ELBO and ImageReward on 5 of them, similar performance on 1, and slight degradation in 1. A representative example of the improvement is shown in Fig. 4. Average metrics are reported in Table 3. For further details see Appendices D.4 and E.

Table 3: FFHQ text-conditional sampling results.

| | ELBO ($\uparrow$) | $\mathbb{E}[\log r(\mathbf{x}, \mathbf{y})]$ ($\uparrow$) | CLIP Diversity ($\uparrow$) |
|---|---|---|---|
| Prior | $-117.0$ | $-1.17$ | $0.36$ |
| $TB_\theta$ (fixed var.) | $98.8$ | $1.42$ | $0.24$ |
| $TB_{\theta,\varphi}$ | $104.5$ | $1.49$ | $0.24$ |

$TB_\theta$ (fixed var.) (Venkatraman et al., 2025)

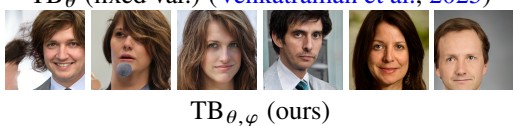

$TB_{\theta,\varphi}$ (ours)

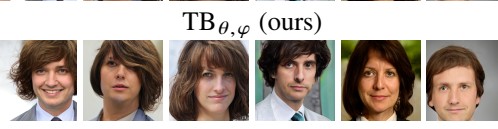

Figure 4: Decoded latents sampled with the same random seeds from outsourced diffusion samplers trained with a StyleGAN3 prior and ImageReward with prompt 'A person with medium length hair'.

## 5 CONCLUSION

In this paper we present the benefits of using learnable variance and learnable destruction process in diffusion samplers. We empirically find that these modifications help to more accurately model complex energy landscapes, especially with few sampling steps. We also contribute to the understanding of training diffusion samplers by studying techniques that improve their stability and convergence speed. We hope that these results inspire the community to scale our findings to other distributions and domains. Another interesting direction for future work would be to rigorously study the optimal parametrisations of the generation and destruction processes – including non-Gaussian transitions – and the theoretical limits of sampling with discrete-time learnt diffusions.

**Discussion of limitations.**    While the proposed method for training the destruction process demonstrates applicability across a series of established benchmarks and shows potential scalability in our StyleGAN3 experiment, it introduces additional hyperparameters, with the destruction learning rate being particularly important. This adds some complexity when applying the method to new settings. The use of target networks also incurs extra training-time overhead, although this can be partially offset by the ability to use fewer discretization steps. At inference time, however, our sampler does not introduce any additional computational cost compared to standard samplers.

## REPRODUCIBILITY STATEMENT

Algorithmic details for stability techniques are presented in Appendix C. Detailed descriptions of target distributions used in experiments are presented in Appendix D.1. All experimental details needed to reproduce the results are provided in Appendix D.3 and Appendix D.4. In addition, the source code with installation and running instructions is attached in the supplementary materials of the submission.

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

## A    RELATED WORK

**Diffusion samplers.**    While diffusion models were first introduced with the goal of approximating a distribution from which samples are available by an iterative denoising process (Sohl-Dickstein et al., 2015; Ho et al., 2020; Song et al., 2021), *diffusion samplers* – on which the modern line of work started with Vargas et al. (2022); Zhang & Chen (2022); Vargas et al. (2023) and continues to rapidly expand (Vargas et al., 2024; Richter & Berner, 2024; Akhound-Sadegh et al., 2024; Sendera et al., 2024; Blessing et al., 2025a; Havens et al., 2025, *inter alia*) – seek to amortise the cost of sampling from a given target density $p_{\text{target}}$, which can be queried for the energy and possibly its gradient, by training a generation process to approximate it. Algorithms such as PIS (Zhang & Chen, 2022) and DDS (Vargas et al., 2023) achieve this by minimising the KL divergence between destruction and generation processes. Other divergences have been considered for their better numerical properties, for example, Richter & Berner (2024) uses a second-moment divergence (Richter et al., 2020). In Lahlou et al. (2023); Zhang et al. (2023b), diffusion samplers are understood as a generalised instance of GFlowNets (Bengio et al., 2021), which are deep reinforcement learning algorithms that allow for using exploration techniques and off-policy training and do not require access to the gradients of $p_{\text{target}}$ (although physics-informed parametrisations may use this gradient). Subsequent work (Zhang et al., 2024; Sendera et al., 2024; Kim et al., 2025b) has applied training objectives from GFlowNet literature (Malkin et al., 2022; Madan et al., 2023; Pan et al., 2023) to the diffusion sampling case and explored the benefits of the flexible off-policy training that GFlowNets allow (Sendera et al., 2024; Phillips & Cipcigan, 2024; Kim et al., 2025b). Finally, (Berner et al., 2025) builds theoretical connections among various diffusion sampling objectives and their continuous-time limits.

**Trainable destruction processes in diffusion samplers.**    Previous contributions, such as (Vargas et al., 2024; Richter & Berner, 2024; Blessing et al., 2025a;b), considered a more restricted setting for trainable destruction processes. Below we discuss these papers in more details.

The authors in Blessing et al. (2025a;b) considered learnable drift for the destruction process. At the same time, both works consider a fixed variance for both generation and destruction processes, while we demonstrate great improvement with a learnable variance[5]. Note also that both papers rely on a parametrisation that includes the energy function gradient $\nabla_x \mathcal{E}(x)$, which might be computationally expensive or unavailable in some applications. We also note that Blessing et al. (2025a) considers a different setting of underdamped Langevin dynamics. Consequently, it is not fully clear whether it is the learnable destruction drift or the augmentation of the space by momentum variables improves performance. The authors in Richter & Berner (2024) also performed a small-scale experiment with learnable drift (and fixed variance) of the destruction process, but observed limited improvements and training instabilities with the utilised parameterisation. Similarly to Blessing et al. (2025a;b), the authors in Richter & Berner (2024) did not provide a systematic ablation study for this approach.

Vargas et al. (2024) introduces Controlled Monte Carlo Diffusions (CMCD), a sampling framework built on paired generation and destruction diffusion processes. A core principle of CMCD is that the generation and destruction drifts are not modeled independently. Instead, they are deterministically linked by a single learnt control function, a constraint justified by Nelson's relation for time-reversal. Our work deviates from this theoretically rigid coupling in favor of greater practical flexibility. We relax the time-reversal constraint and predict the generation and destruction drifts independently, as well as introduce an additional degree of freedom by learning the variance.

**GFlowNets.**    Generative flow networks (GFlowNets; Bengio et al., 2021; 2023) have been introduced as a general framework for sampling from distributions by solving a sequential decision-making problem, training a stochastic policy to construct objects step by step. GFlowNets represent a synthesis of variational inference and reinforcement learning (RL) paradigms (Malkin et al., 2023; Zimmermann et al., 2023; Tiapkin et al., 2024; Deleu et al., 2024). Beyond their original application in biological structure discovery (*e.g.*, Jain et al., 2022; Shen et al., 2024; Cretu et al., 2025), GFlowNets have found uses in probabilistic inference over symbolic latent variables (Deleu et al., 2022; van Krieken et al., 2023; Hu et al., 2023; Deleu et al., 2023; Zhou et al., 2024; Silva et al., 2024), combinatorial optimisation (Zhang et al., 2023a;c; Kim et al., 2025a), and reasoning or planning in language (Hu et al., 2024; Song et al., 2024; Lee et al., 2025; Younsi et al., 2025). Although

---

[5]Blessing et al. (2025a) presented an experiment with learnable variance with time- and state-independent parametrisation for underdamped processes (learnable diagonal matrix), yet they did not report any improvements out of it

initially defined for discrete spaces, the framework can be extended to continuous domains Lahlou et al. (2023) and non-acyclic state spaces (*i.e.*, nonconstrutive actions) (Brunswic et al., 2024; Morozov et al., 2025). GFlowNets approach the training of amortised samplers from a RL perspective and correspondingly require solving the RL challenges of exploration (Rector-Brooks et al., 2023; Sendera et al., 2024; Phillips & Cipcigan, 2024; Madan et al., 2025; Kim et al., 2025b), credit assignment (Malkin et al., 2022; Pan et al., 2023; Jang et al., 2024b), and generalisation (Silva et al., 2025; Atanackovic & Bengio, 2025). Relevant to our work, design and training of the destruction process in *discrete* problems is a focus of Malkin et al. (2023); Mohammadpour et al. (2024); Jang et al. (2024a); Gritsaev et al. (2025), where learning the destruction process is shown to improve convergence and mode discovery.

**Learning the destruction process in diffusion models trained on data.** There is a line of work (Bartosh et al., 2024a;b; Sahoo et al., 2024; Nielsen et al., 2024) that studies learning the destruction process (called the 'forward' process, clashing with the opposite convention in place for diffusion samplers) for diffusion models in image domains, instead of using a fixed one, such as VP or VE SDE (Song et al., 2021). This results in a broader class of models and is shown to improve log-likelihood and visual quality of the generated samples, as well as reducing the required number of training steps. Objectives inspired by GFlowNets were also applied to this problem in Lahlou et al. (2023); Zhang et al. (2023b). Learning the destruction process for diffusion models was also studied in discrete domains, *e.g.*, Kong et al. (2023) and Wang et al. (2025) learn the order in which nodes are removed from a graph by the destruction process. However, there is a clear distinction between diffusion models that are trained on a dataset of samples and diffusion samplers, that have access to the energy function of the target distribution during training, but no access to actual samples from this distribution. Because of this, training objectives used in these two settings are fundamentally different as they use different forms of ground truth data. Thus, methods for learning destruction processes in diffusion models are not applicable to our setting; conversely, the methods we develop do not apply to the setting of diffusion models trained on data.

Our work is also adjacent to the **Schrödinger bridge (SB) problem**. In continuous time, the learning of destruction and generation processes that stochastically transport a source distribution $p_0$ to $p_{\text{target}}$ is known as a *bridge problem*. The SB problem seeks the unique bridge that is closest to some reference process and can be solved by algorithms based on iterative proportional fitting in a time discretisation (IPF; Vargas et al., 2021; De Bortoli et al., 2021) or various other methods (Chen et al., 2022; Stromme, 2023; Shi et al., 2023; Tong et al., 2024). Unlike diffusion samplers, these algorithms typically assume samples from $p_{\text{target}}$ are available. Many of these algorithms, when typical approximations are made in training, at convergence yield bridges that are not necessarily the SB. In this work, we constrain the destruction process to be a bridge from $p_{\text{target}}$ to $p_0$, meaning that in the continuous-time limit iterative proportional fitting would converge in a single step. However, the discrete-time TLM update for the destruction process (§3.2), which trains $\overleftarrow{p}_\varphi$ on trajectories sampled from $\overrightarrow{p}_\theta$, is in fact equivalent to an (unconverged) maximum-likelihood IPF step as in Vargas et al. (2021).

# B MORE ON DIFFUSION SAMPLERS

## B.1 ON KL DIVERGENCE BETWEEN PROCESSES WITH DIFFERENT VARIANCES.

If two path space measures $\mathbb{P}_1$ and $\mathbb{P}_2$ defined by SDEs have different diffusion coefficients $g_1(t)$ and $g_2(t)$, they are not absolutely continuous with respect to each other, therefore, $\text{KL}(\mathbb{P}_1 \| \mathbb{P}_2) = \infty$. To see this, notice that the quadratic variation of $\mathbb{P}_i$ on the time interval $[t, t']$ is almost surely $\int_t^{t'} g_i(s)^2 \mathrm{d}s$. Therefore, if $\mathbb{P}_1 \ll \mathbb{P}_2$, then the $\int_t^{t'} g_1(s)^2 \mathrm{d}s = \int_t^{t'} g_2(s)^2 \mathrm{d}s$ for every $t < t'$, which implies $g_1 = g_2$ if both are continuous.

## B.2 ON SOFT RL EQUIVALENCE.

In this section, we show that training a diffusion sampler with a finite number of steps and a fixed destruction process is equivalent to solving an entropy-regularized reinforcement learning (RL) problem.

We formalize the RL problem using a finite-horizon Markov Decision Process (MDP) (Puterman, 2014, Chapter 4), defined as the tuple $\mathcal{M} = (\mathcal{S}, \mathcal{A}, \mathsf{P}, \mathsf{r}, H, s_0)$, where $\mathcal{S}$ and $\mathcal{A}$ are measurable state and action spaces, $\mathsf{P}_h(\mathrm{d}s'|s, a)$ is a time-inhomogeneous transition kernel, $\mathsf{r}_h(s, a)$ is a time-

dependent reward function with terminal reward $r_H(s)$, $H$ is the planning horizon, and $s_0$ is the initial state. We focus on deterministic MDPs, where the transition kernel is

$$P_h(ds' \mid s, a) = \delta_{T_h(s,a)}(ds'),$$

with $T_h(s, a)$ a deterministic transition map and $\delta_y(dx)$ the Dirac measure at $y$. The action space is equipped with a base measure $d\mu(a)$ (*e.g.*, the Lebesgue measure).

A time-inhomogeneous policy $\pi = \{\pi_h\}_{h=0}^{H-1}$ is a collection of conditional densities $\pi_h(a|s)$ with respect to $d\mu(a)$. The corresponding entropy-regularized (or soft) value function (Neu et al., 2017; Geist et al., 2019) is defined as

$$V_{\lambda,0}^\pi(s) = \mathbb{E}_\pi \left[ \sum_{h=0}^{H-1} (r_h(S_h, A_h) - \lambda \log \pi_h(A_h|S_h)) + r_H(S_H) \,\middle|\, S_0 = s \right],$$

where $\lambda \geq 0$ is a regularization coefficient, and the expectation is over the trajectory induced by $A_h \sim \pi_h(\cdot|S_h)$ and $S_{h+1} = T_h(S_h, A_h)$ for $h = 0, \ldots, H-1$.

We call a destruction process $\overleftarrow{p} = (\overleftarrow{p_t})_{t=0,\Delta t,\ldots,1}$ *regular* if $\overleftarrow{p_{\Delta t}}(dx_0|x_{\Delta t}) = \delta_0(dx_0)$ and $\overleftarrow{p_t}(\cdot|x_t)$ has a full support of $\mathbb{R}^d$ for all $t > \Delta t$. We use $\overleftarrow{p_{t+\Delta t}}(x_t|x_{t+\Delta t})$ as a corresponding density.

**Theorem B.1.** *Define $\mathcal{E}(x) \colon \mathbb{R}^d \to \mathbb{R}$ as an energy function, a target density $p_{\text{target}}(x) = \exp(-\mathcal{E}(x))/Z$, and a regular destruction process $\overleftarrow{p} = (\overleftarrow{p_t})_{t=0,\Delta t,\ldots,1}$.*

*Define a Markov decision process $\mathcal{M}_{\text{DS}}$ with a state space equal to $\mathcal{S} = \mathbb{R}^d$, an action space equal to $\mathcal{A} = \mathbb{R}^d$, deterministic transition kernel defined by the following transition function $T_h(s, a) = a$, a planning horizon $H = T$, and reward function $r_h(s, a) = \log \overleftarrow{p}_{h \cdot \Delta t}(a \mid s)$ for $h < H$, with terminal reward $r_H(x) = -\mathcal{E}(x)$.*

*Then, for initial state $s_0 = 0$, the soft value with $\lambda = 1$ in the MDP $\mathcal{M}_{\text{DS}}$ satisfies*

$$V_{\lambda=1,0}^\pi(s_0) = \log Z - \mathbb{D}_{\text{KL}}(p_0 \overrightarrow{p_\pi} \| p_{\text{target}} \overleftarrow{p}),$$

*where $\overrightarrow{p_\pi}$ is a generation process that corresponds to a policy $\pi$: $\overrightarrow{p_{\pi,t}}(x_{t+\Delta t} \mid x_t) = \pi_{t/\Delta t}(x_{t+\Delta t} \mid x_t)$. As a result, the policy corresponding to the optimal policy $\pi_h^\star$ in the entropy-regularized MDP with $\lambda = 1$ provides a generation process that samples from the target distribution $p_{\text{target}}(x) \propto \exp(-\mathcal{E}(x))$.*

*Proof.* Let us start from the expression (13) for the reverse KL divergence between the corresponding generation and destruction processes:

$$\mathbb{D}_{\text{KL}}(p_0 \overrightarrow{p_\pi} \| p_{\text{target}} \overleftarrow{p}) = \mathbb{E}_{X_{0,\Delta t,\ldots,1} \sim \overrightarrow{p_\pi}(X_{0,\Delta t,\ldots,1})} \left[ \log \frac{p_0(X_0) \overrightarrow{p_\pi}(X_{\Delta t\ldots,1} \mid X_0)}{\exp(-\mathcal{E}(X_1)) \overleftarrow{p}(X_{0,\ldots,(T-1)\Delta t} \mid X_1)} \right] + \log Z.$$

Next, we use a chain rule for the probability distribution to study the first term in the expansion above:

$$\mathbb{E}_{X_{0,\Delta t,\ldots,1} \sim \overrightarrow{p_\pi}(X_{0,\Delta t\ldots,1})} \left[ \sum_{i=0}^{T-1} \log \overrightarrow{p_{\pi,i\Delta t}}(X_{(i+1)\Delta t} \mid X_{i\Delta t}) - \log \overleftarrow{p_{(i+1)\Delta t}}(X_{i\Delta t} \mid X_{(i+1)\Delta t}) + \mathcal{E}(X_1) \right].$$

To show the equivalence with the corresponding soft RL definition of the value, we rename variables as follows: $T \mapsto H$, $i \mapsto h$, and $S_{h+1} = A_h = X_{(i+1)\Delta t}$, apply a definition of a reward function and of the generation process induced by policy:

$$\mathbb{D}_{\text{KL}}(p_0 \overrightarrow{p_\pi} \| p_{\text{target}} \overleftarrow{p}) - \log Z = \mathbb{E}_\pi \left[ \sum_{h=0}^{H-1} (\log \pi_h(A_h|S_h) - r_h(S_h, A_h)) - r_H(S_H) \right].$$

In the right-hand side of the equation we see exactly a negative value function with $\lambda = 1$. $\qquad \square$

The same result was obtained by Tiapkin et al. (2024) in the GFlowNet framework. In addition, we would like to add that this perspective allows to treat the Trajectory Balance loss as a specific instance of Path Consistency Learning Nachum et al. (2017), as it was shown for GFlowNets in Deleu et al. (2024) (see also Uehara et al. (2024) for a discussion in the context of RL-based fine-tuning of diffusion models).

## C TECHNIQUES FOR STABILITY

In this section, we describe the implementation details of stability techniques and discuss unsuccessful approaches for destruction process training.

**Shared backbone.** Our architecture is built upon the one used in Sendera et al. (2024), uses time and state encoders, a shared backbone and 2 different final layers for generation and destruction policy modelling. We use an MLP with GELU (Hendrycks & Gimpel, 2016) activation for the shared backbone. For experiments on Manywell, Distorted Manywell, and GAN, this MLP has 4 layers, and for other energies it has 2.

**Separate optimisers.** The optimiser for generation policy updates parameters of time and state encoders, the backbone, and the final layer modelling the generation process. The optimiser for destruction policy updates the parameters of the time and state encoders, the backbone, and the final layer modelling the destruction process. For both optimisers, we use Adam (Kingma & Ba, 2015) with standard parameters and weight decay of $10^{-7}$.

**Target network.** Using a target network introduces a specific coefficient $\tau$, which defines the update speed of the target network. On each iteration, the target weights are updated by:

$$\overline{\theta} = (1 - \tau)\overline{\theta} + \tau\theta, \tag{17}$$

where $\overline{\theta}$ are the target network weights. If $\tau$ is set to 0, the target network is the current policy network, while higher $\tau$ leads to a slower evolution of the target network. For the generation policy network, the second moment loss in (14) transforms into:

$$\mathbb{D}_{\tilde{p}}^{\text{gen.}}(p_0 \overrightarrow{p_\theta} \| p_{\text{target}} \overleftarrow{p_\varphi}) = \mathbb{E}_{X_{0,\Delta t,\dots,1} \sim \tilde{p}(X_{0,\Delta t,\dots,1})} \left[ \log \frac{p_0(X_0)\overrightarrow{p_\theta}(X_{\Delta t\dots,1} \mid X_0)}{\exp(-\mathcal{E}(X_1))\overleftarrow{p_{\overline{\varphi}}}(X_{0,\dots,(T-1)\Delta t} \mid X_1)} + \log \hat{Z} \right]^2, \tag{18}$$

where $\overline{\varphi}$ is the frozen destruction target network weights, and other variables are the same as in (14).

Similarly, the second moment loss for the destruction policy transforms into:

$$\mathbb{D}_{\tilde{p}}^{\text{destr.}}(p_0 \overrightarrow{p_\theta} \| p_{\text{target}} \overleftarrow{p_\varphi}) = \mathbb{E}_{X_{0,\Delta t,\dots,1} \sim \tilde{p}(X_{0,\Delta t,\dots,1})} \left[ \log \frac{p_0(X_0)\overrightarrow{p_{\overline{\theta}}}(X_{\Delta t\dots,1} \mid X_0)}{\exp(-\mathcal{E}(X_1))\overleftarrow{p_\varphi}(X_{0,\dots,(T-1)\Delta t} \mid X_1)} + \log \hat{Z} \right]^2, \tag{19}$$

where $\overline{\theta}$ is the frozen generation target generation network, and other variables are as in (14).

**Prioritised experience replay.** We use the implementation of PER (Schaul et al., 2016) from `torchrl` library (Bou et al., 2023). We set the temperature parameter $\alpha$ to 1.0 and the importance sampling correction coefficient to 0.1 (similar parameter values were used in Tiapkin et al. (2024)).

**Better exploration in off-policy methods.** We use the existing techniques proposed by Lahlou et al. (2023); Sendera et al. (2024) to facilitate exploration during training. We use a replay buffer of terminal states updated by Langevin dynamics (as studied by Sendera et al. (2024)), that is used to sample trajectories for training via the destruction process. We also sample trajectories from the current generation policy, but with increased variance on each step, with the added variance annealed to zero over the first 10 000 iterations (similar to the techniques studied by Malkin et al. (2023); Lahlou et al. (2023)).

**Other considerations.** In addition to the techniques discussed in §3.3 we also present design choices that we tried in our experiments, but which turned out to be unsuccessful. We share them to offer deeper intuition behind the development of our final methodology:

- **Different parametrisation.** We tried to predict destruction variance in the log-scale in the same way as the generation variance (11):

$$\beta'_\varphi(X_t, t) = \exp\left\{ C_1 \tanh\left( \text{NN}_\theta^{(2)}(X_t, t) \right) \right\}, \tag{20}$$

  but this parametrisation caused rapid fluctuations in the destruction policy and convergence was dramatically slower than with the parametrisation in (12).
- **Linear annealing.** In experiments on synthetic tasks, we set the constant $C_2$ in (12) to 0.9. We also tried to increase $C_2$ linearly from 0 to 0.9 over the duration of training. We initially thought this to be efficient since both generation and destruction processes are less stable in the beginning of training when modes are unexplored.

# D EXPERIMENT DETAILS

## D.1 DEFINITION OF ENERGIES

We present contour levels of 2-dimensional Gaussian mixtures in Figure 5.

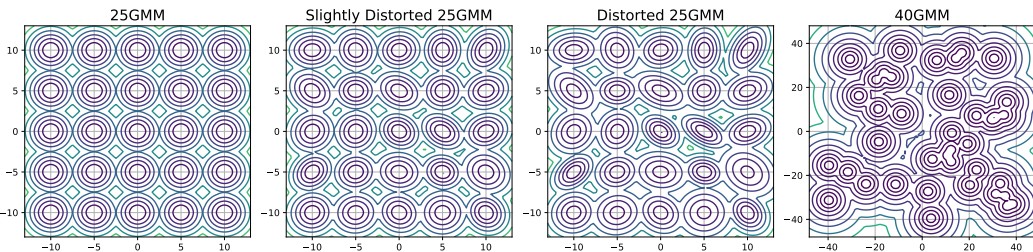

Figure 5: Contour levels of 2-dimensional Gaussian mixture densities used in the experiments.

**25GMM.** This is a mixture of 25 different Gaussians in a 2-dimensional space. Each component is Gaussian with variance 0.3. The means are arranged on a grid given by the Cartesian product $\{-10, -5, 0, 5, 10\} \times \{-10, -5, 0, 5, 10\}$.

**Slightly Distorted 25GMM.** This environment is obtained by a slight modification of 25GMM. The means are located in the same positions, but we distort the initial variance matrices with the following rule:

$$C_i = \left( \begin{bmatrix} \sqrt{0.3} & 0 \\ 0 & \sqrt{0.3} \end{bmatrix} + d \begin{bmatrix} \xi_{i1} & \xi_{i2} \\ \xi_{i3} & \xi_{i4} \end{bmatrix} \right)^\top \cdot \left( \begin{bmatrix} \sqrt{0.3} & 0 \\ 0 & \sqrt{0.3} \end{bmatrix} + d \begin{bmatrix} \xi_{i1} & \xi_{i2} \\ \xi_{i3} & \xi_{i4} \end{bmatrix} \right), \tag{21}$$

where $C_i$ is the covariance matrix of the $i$-th mode, $\xi_{ij} \sim \mathcal{N}(0, 1)$, and $d$ is set to 0.05. To achieve a fair comparison, we sample the random variables with a predefined seed 42, hence all algorithms are compared on the same distribution.

**Distorted 25GMM.** The same as Slightly Distorted 25GMM, but with $d$ equal 0.1.

**125GMM.** The same as 25GMM, but in $\mathbb{R}^3$ with means at $\{-10, -5, 0, 5, 10\}^3$.

**40GMM.** This distribution is taken from Midgley et al. (2023). It consists of 40 equally weighted mixture components with components sampled as:

$$\pi_k(x) = \mathcal{N}(x; \mu_k, I)$$
$$\mu_k \sim \mathcal{U}(-40, 40).$$

**Easy/Hard Funnel.** The funnel distribution serves as a classical benchmark in the evaluation of sampling methods. It is defined over a ten-dimensional space, where the first variable, $x_0$, is drawn from a normal distribution with mean 0 and variance 1 (Easy Funnel) or 9 (Hard Funnel), $x_0 \sim \mathcal{N}(0, 1)$. Given $x_0$, the remaining components $x_{1:9}$ follow a multivariate normal distribution with mean vector zero and covariance matrix $\exp(x_0)I$, where $I$ denotes the identity matrix. This conditional relationship is expressed as $x_{1:9} \mid x_0 \sim \mathcal{N}(0, \exp(x_0)I)$.

**Manywell.** The distribution is defined over a 32-dimensional space and is constructed as the product of 16 identical 2-dimensional double-well distributions. Each of these two-dimensional components is governed by a potential function, $\mu(x_1, x_2)$, given by $\mu(x_1, x_2) = \exp\left(-x_1^4 + 6x_1^2 + 0.5x_1 - 0.5x_2^2\right)$.

**Distorted Manywell.** We modify the Manywell potential function for each 2-dimensional components:

$$\mu(x_{2i-1}, x_{2i}) = \exp\left(-a_{i,1}x_{2i-1}^4 + 6a_{i,2}x_{2i-1}^2 + 0.5a_{i,3}x_{2i-1} - 0.5a_{i,4}x_{2i}^2\right),$$

where $a_{i,j} \sim \mathcal{U}(0.75, 1.25)$.

**VAE** This problem is taken from Sendera et al. (2024). The task is to sample from a 20-dimensional latent posterior, defined as $p(z|x) \propto p(z)p(x|z)$, where $p(z)$ is a fixed prior and $p(x|z)$ is a pretrained VAE decoder, using a conditional sampler $q(z|x)$ conditioned on the input image $x$.

## D.2 METRICS

The ELBO and EUBO metrics are defined as follows:

$$\text{ELBO} = \mathbb{E}_{X_{0,\dots,1} \sim p_0(X_0)\overrightarrow{p_\theta}(X_{0,\Delta t \dots,1})} \left[ \log \frac{\exp(-\mathcal{E}(X_1))\overleftarrow{p_\varphi}(X_{0,\dots,(T-1)\Delta t} \mid X_1)}{p_0(X_0)\overrightarrow{p_\theta}(X_{\Delta t \dots,1} \mid X_0)} \right]$$

$$\text{EUBO} = \mathbb{E}_{X_{0,\dots,1} \sim p_{\text{target}}(X_1)p_\varphi(X_{0,\dots,(T-1)\Delta t}\mid X_1)} \left[ \log \frac{\exp(-\mathcal{E}(X_1))\overleftarrow{p_\varphi}(X_{0,\dots,(T-1)\Delta t} \mid X_1)}{p_0(X_0)\overrightarrow{p_\theta}(X_{\Delta t \dots,1} \mid X_0)} \right]$$

Both expectations are estimated using 2048 Monte Carlo samples. For the 2-Wasserstein distance $(W_2^2)$, we also use 2048 ground truth and 2048 generated samples.

## D.3 STANDARD BENCHMARKS TRAINING DETAILS

Here we describe the chosen hyperparameters for the final results across synthetic tasks in §4.1. We train all samplers for 25 000 iterations. The diffusion rate $\sigma^2$ is set to 5 for experiments with Gaussian mixtures and to 1 for other energies. The batch size is 512 for all tasks except VAE. We apply zero initialisation for the final layers of the neural networks to obtain a uniform output in the beginning of the training. Additionally, we perform clipping on the output of the policy network by $10^{-4}$. We also apply gradient clipping with value 200. The target neural network update speed $\tau$ is set 0.05.

We set $C_1$ in (11) and $C_2$ in (12) to 4.0 and 0.9 respectively.

For all experiments, we set $\text{lr}_\theta$ (learning rate for generation policy neural network) to $10^{-3}$. The normalisation constant $\log Z$ is trained with learning rate $10^{-1}$. We tune $\text{lr}_\varphi$ (learning rate for destruction policy neural network) specifically for each energy and find that optimal $\text{lr}_\varphi$ is always less than or equal to $\text{lr}_\theta$. For the samplers trained with TB loss, we set $\text{lr}_\varphi$ to $\text{lr}_\theta$ in experiments with 25GMM and 125GMM, choose the optimal $\text{lr}_\varphi$ between $\{\text{lr}_\theta, 10^{-1} \times \text{lr}_\theta\}$ for 40GMM, set $\text{lr}_\varphi$ to $\text{lr}_\theta$ for Easy Funnel and to $10^{-3} \times \text{lr}_\theta$ for Hard Funnel, we set $\text{lr}_\varphi$ to $10^{-5} \times \text{lr}_\theta$ for $T = 5$ and to $10^{-4} \times \text{lr}_\theta$ for $T = 10$ and $T = 20$ in Manywell and Distorted Manywell. For PIS, we choose optimal $\text{lr}_\varphi$ between $\{\text{lr}_\theta, 10^{-1} \times \text{lr}_\theta, 10^{-2} \times \text{lr}_\theta\}$ in experiments with Gaussian mixtures. We use an exponential learning rate schedule, which multiplies learning rate by $\gamma$ after each on-policy gradient step. We set $\gamma$ to 0.9999 in all experiments.

For off policy TB, we set the replay ratio to 3. We set the size of the experience replay buffer to 500000 and the size of the buffer used in local search to 600 000. All hyperparameters for local search are taken from Sendera et al. (2024). We use exploration factor of 0.3 for Gaussian mixtures, 0.2 for Easy Funnel and Hard Funnel, and 0.1 for Manywell and Distorted Manywell. We note that we use replay buffers and off-policy exploration in all methods and configurations that allow for off-policy training to ensure a fair comparison.

We consider the architecture from Sendera et al. (2024), but we stack the time encoding with the state encoding rather than summing them. We use 2 joint layers for all tasks with the except for 40GMM, Manywell and Distorted Manywell, where we use 4 joint layers. We set state, time, and hidden dimension sizes to 64 across all environments.

We use a harmonic discretisation scheme for all mixtures of Gaussians since this time discretisation leads to better sampling quality compared to uniform. It partitions the time space unevenly, making steps large around $t = 0$ and smaller closer to $t = 1$. The code for harmonic discretisation is presented in Appendix D.3. For other energies, we apply uniform time discretization.

```
def harmonic_discretizer(batch_size: int, trajectory_length: int):
    step_sizes = 1 / arange(1, trajectory_length + 1)
    sum_step_sizes = sum(step_sizes)
    step_proportions = step_sizes / sum_step_sizes
    split_points = cumsum(step_proportions)
    return concatenate([0.0, split_points])
```

In the VAE experiment, VarGrad requires adaptation for the conditional setting because the partition function depends on the conditioning image x. This necessitates computing variance over multiple trajectories that share the same condition. We implement this by sampling 5 trajectories for each condition within a minibatch and computing the VarGrad loss on each set of 5. Thus, the batch size in this task is set to $5 \times 512$. All other settings are taken from Sendera et al. (2024).

### D.4 GAN LATENT SAMPLER TRAINING DETAILS

We use similar setup and hyperparameters to the ones described in Appendix D.3, with a number of differences. We use smaller values of $C_1 = 1.0$ and $C_2 = 0.1$, which we found to improve the training stability and sampling quality in this task. We also use a smaller batch size of 128 and larger hidden size of 1024 for the MLP network (note that Venkatraman et al. (2025) uses a variant of UNet architecture (Ronneberger et al., 2015), while we use a variant of the architecture from Sendera et al. (2024), see Appendix D.3). We set $\sigma^2 = 1$, which matches the variance of the prior. We set $\mathrm{lr}_\theta = 10^{-3}$, $\mathrm{lr}_\varphi = 0.2 \times \mathrm{lr}_\theta$, $\gamma = 0.9999$. The exploration factor is set to 0.1 and the replay ratio is set to 6. We found it beneficial to set the target neural network update speed $\tau$ to a higher value of 0.15. We train all samplers for 20 000 iterations. We use the same log-reward scaling $\beta = 100$ as in Venkatraman et al. (2025).

We note that replay buffers and off-policy exploration are used both for the baseline and for our approach to ensure a fair comparison. We do not use the local search method proposed in Sendera et al. (2024) as it requires access to gradients of the target energy function, which would require costly differentiation through the GAN generator and ImageReward. Thus all models considered in this experiment require access only to $\mathcal{E}(z)$.

For GAN latent space sampling experiments we used NVIDIA V100 GPUs. Our implementations are based upon the published code of Sendera et al. (2024), as well as the official implementations of Karras et al. (2021); Radford et al. (2021); Xu et al. (2023).

Extending the results from Table 3, Fig. 6 depicts metric differences across all prompts utilized in this experiment.

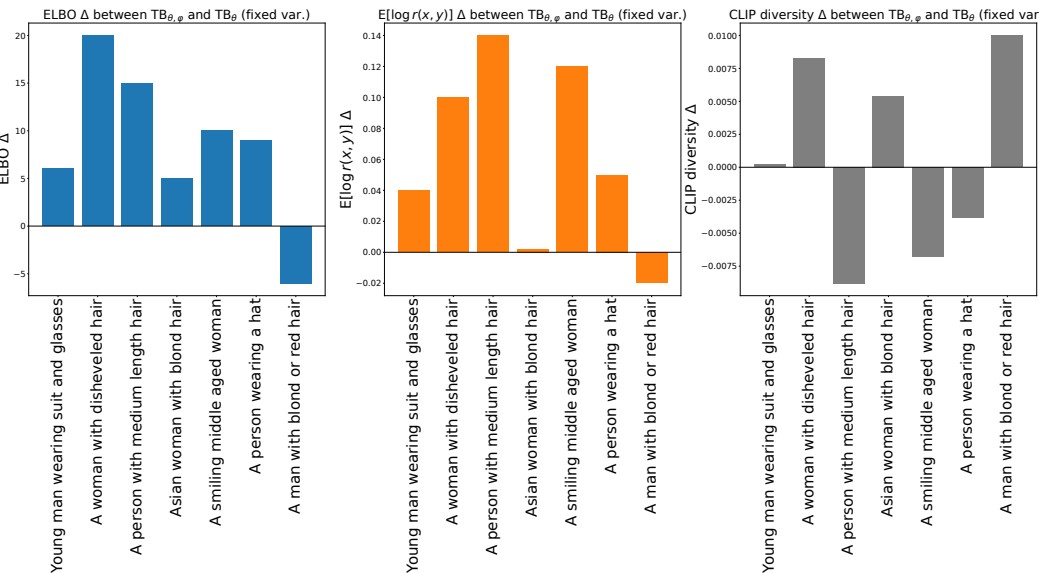

Figure 6: FFHQ text-conditional latent sampling results across different prompts. The figure depicts the difference in metrics between $\mathrm{TB}_{\theta,\varphi}$ (ours) and $\mathrm{TB}_\theta$ (fixed var.). *Left:* ELBO, *middle:* average ImageReward, *right:* CLIP diversity. Our method improves reward and ELBO in most prompts. Differences in CLIP diversity are minor (less than 0.01 for all prompts), and on average both methods show the same diversity when rounded up to two decimals.

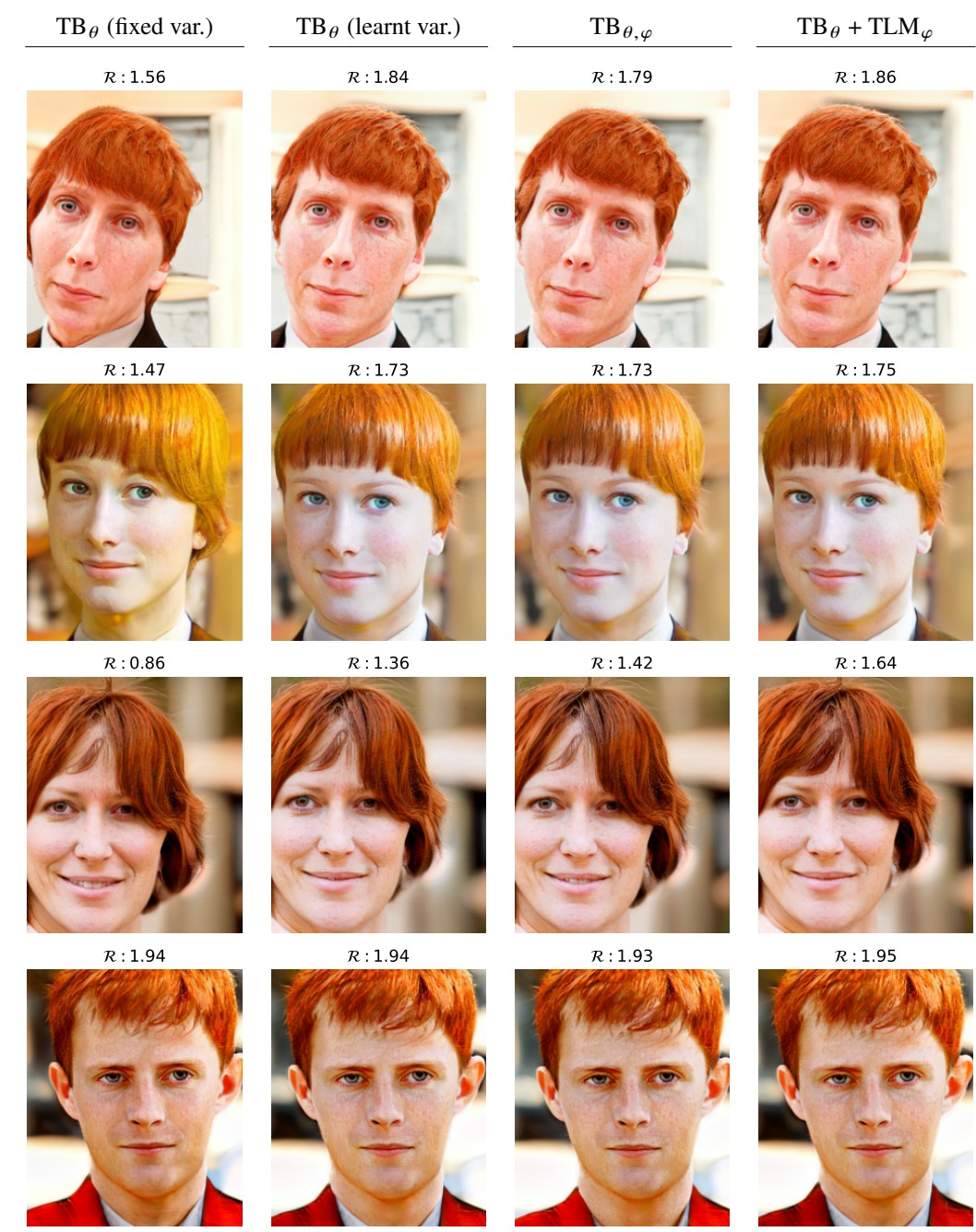

Figure 7: Images generated by StyleGAN3 for the prompt "Person with freckles and short red hair". Each column corresponds to one method, and each row uses the same seed across methods. The ImageReward score for each image is shown in its title.

# E   SUPPLEMENTARY FIGURES

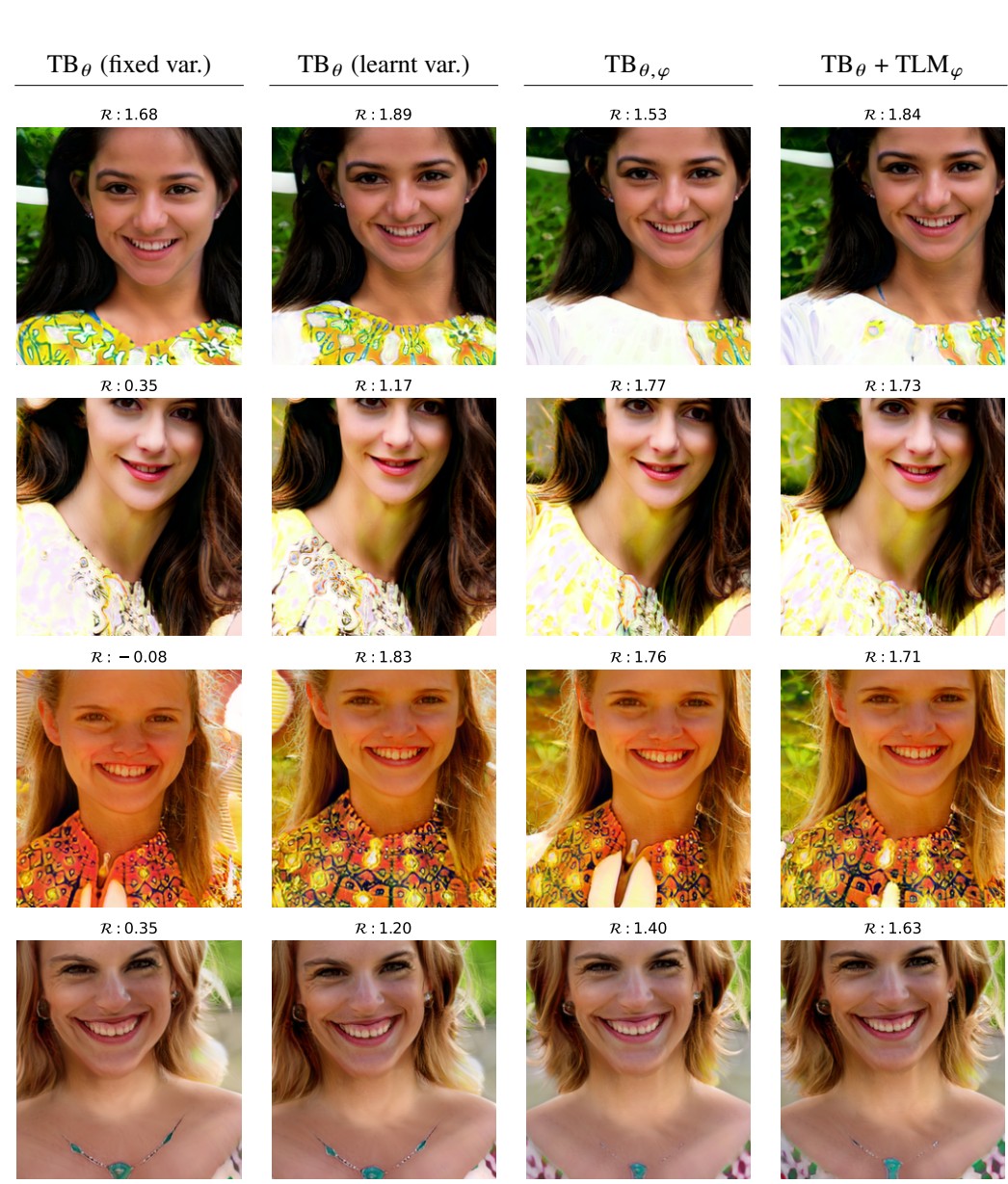

Figure 8: Images generated by StyleGAN3 for the prompt "Smiling woman wearing a summer dress". Each column corresponds to one method, and each row uses the same seed across methods. The ImageReward score for each image is shown in its title.

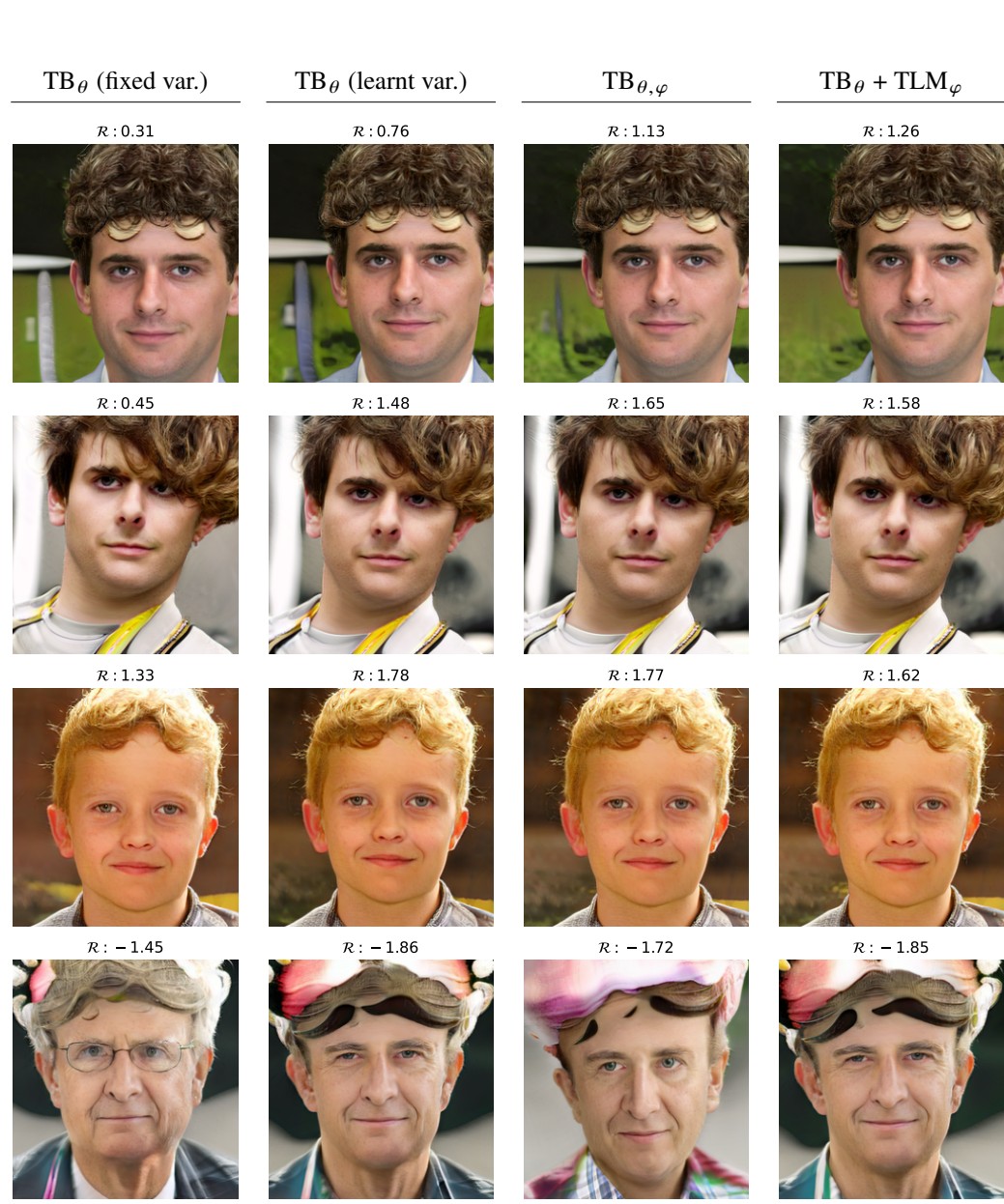

Figure 9: Images generated by StyleGAN3 for the prompt "Person with short curly blond hair". Each column corresponds to one method, and each row uses the same seed across methods. The ImageReward score for each image is shown in its title.

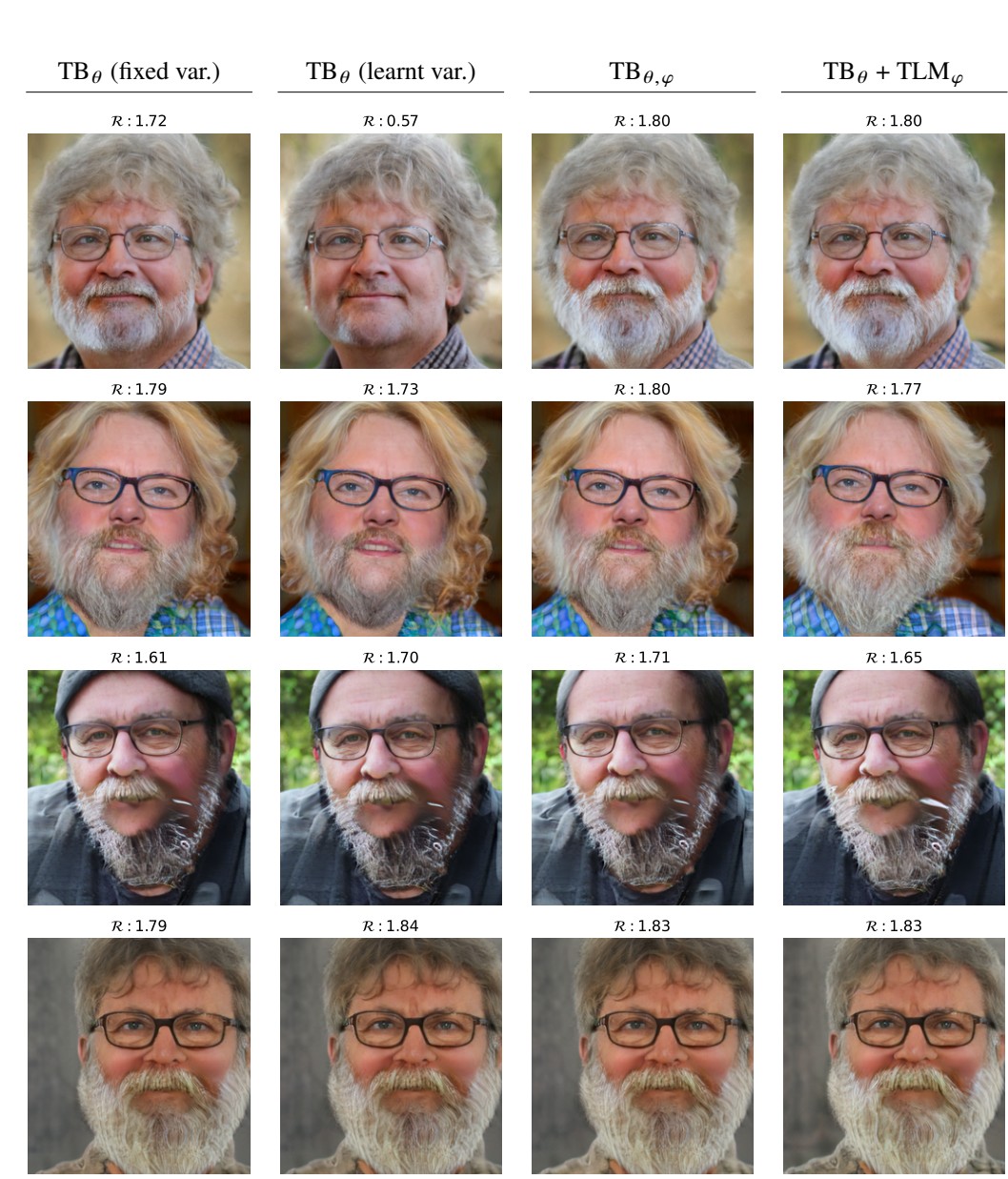

Figure 10: Images generated by StyleGAN3 for the prompt "Middle-aged man with a beard and glasses". Each column corresponds to one method, and each row uses the same seed across methods. The ImageReward score for each image is shown in its title.

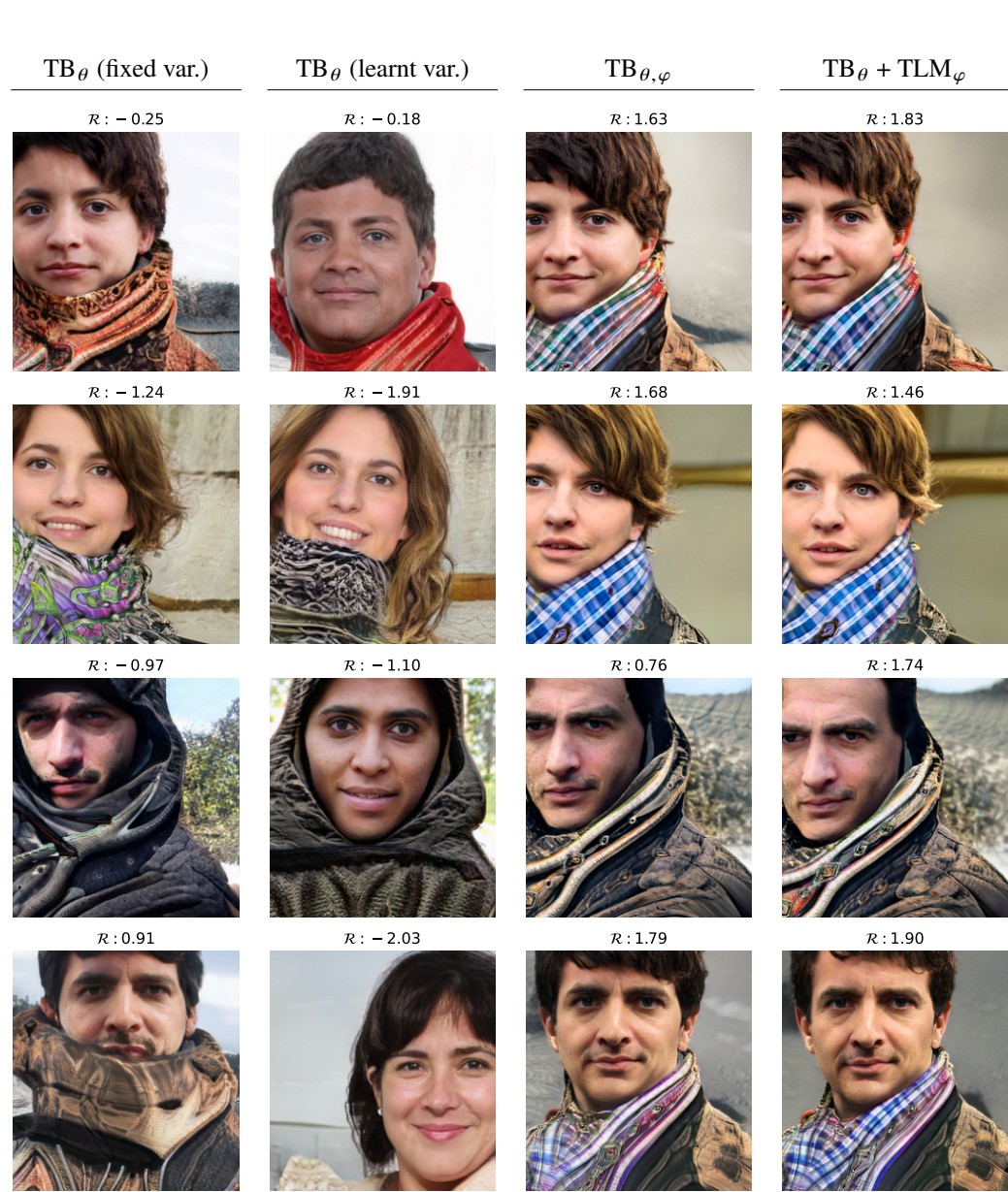

Figure 11: Images generated by StyleGAN3 for the prompt "Man wearing a leather jacket and scarf". Each column corresponds to one method, and each row uses the same seed across methods. The ImageReward score for each image is shown in its title.

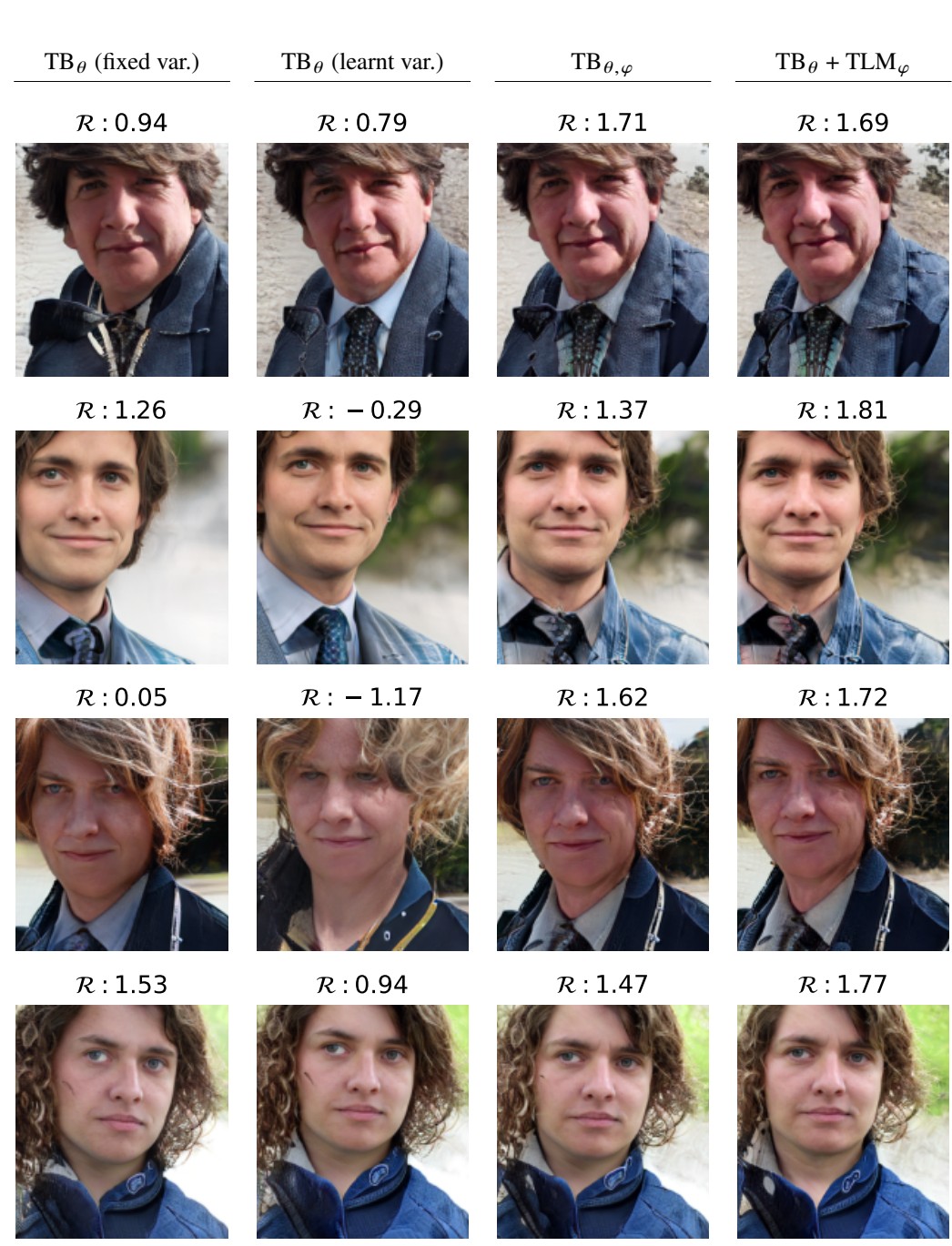

Figure 12: Images generated by StyleGAN3 for the prompt "Person with shoulder-length hair and a denim jacket". Each column corresponds to one method, and each row uses the same seed across methods. The ImageReward score for each image is shown in its title.

## F  SUPPLEMENTARY TABLES

See Tables 4 to 10 on the following pages for full results.

Table 4: Comparison of 4 algorithms by ELBO, EUBO, and 2-Wasserstein between generated and ground truth samples with varying number of discretisation steps on 125GMM. Mean and std over 3 runs are specified.

| | | ELBO (↑) | | | | | |
|---|---|---|---|---|---|---|---|
| Energy ↓ | Method ↓ Steps → | $T = 2$ | $T = 3$ | $T = 4$ | $T = 5$ | $T = 10$ | $T = 20$ |
| | $\text{TB}_\theta$ (fixed var.) | $-7.36_{\pm0.09}$ | $-5.32_{\pm0.01}$ | $-4.29_{\pm0.03}$ | $-3.64_{\pm0.06}$ | $-2.39_{\pm0.03}$ | $-1.72_{\pm0.01}$ |
| | $\text{TB}_\theta + \text{TLM}_\varphi$ (fixed var.) | $-7.33_{\pm0.13}$ | $-5.38_{\pm0.09}$ | $-3.76_{\pm0.15}$ | $-2.88_{\pm0.10}$ | $-1.22_{\pm0.06}$ | $-0.65_{\pm0.04}$ |
| | $\text{TB}_{\theta,\varphi}$ (fixed var.) | $-8.21_{\pm0.13}$ | $-5.51_{\pm0.08}$ | $-3.96_{\pm0.02}$ | $-3.13_{\pm0.05}$ | $-1.29_{\pm0.05}$ | $-0.70_{\pm0.08}$ |
| | $\text{TB}_\theta$ (learnt var.) | $-4.92_{\pm0.16}$ | $-2.27_{\pm0.06}$ | $-1.31_{\pm0.06}$ | $-0.88_{\pm0.08}$ | $-0.36_{\pm0.02}$ | $-0.24_{\pm0.03}$ |
| 125GMM | $\text{TB}_\theta + \text{TLM}_\varphi$ (learnt var.) | $-3.82_{\pm1.11}$ | $-1.67_{\pm0.10}$ | $-1.31_{\pm0.02}$ | $-0.80_{\pm0.10}$ | $-0.34_{\pm0.05}$ | $-0.31_{\pm0.24}$ |
| | $\text{TB}_{\theta,\varphi}$ (learnt var.) | $-5.34_{\pm1.09}$ | $-2.06_{\pm0.06}$ | $-1.31_{\pm0.13}$ | $-0.86_{\pm0.08}$ | $-0.27_{\pm0.09}$ | $-0.06_{\pm0.00}$ |
| | $\text{PIS}_\theta$ (fixed var.) | $-6.94_{\pm0.32}$ | $-5.34_{\pm0.63}$ | $-4.75_{\pm0.73}$ | $-3.68_{\pm0.13}$ | $-4.85_{\pm0.13}$ | $-2.83_{\pm0.12}$ |
| | $\text{PIS}_\theta$ (learnt var.) | $-3.18_{\pm0.59}$ | $-2.75_{\pm0.26}$ | $-2.09_{\pm0.04}$ | $-2.05_{\pm0.04}$ | $-1.87_{\pm0.01}$ | $-2.06_{\pm0.08}$ |
| | $\text{PIS}_\theta + \text{TLM}_\varphi$ (learnt var.) | $-1.06_{\pm0.34}$ | $-1.29_{\pm0.15}$ | $-1.31_{\pm0.13}$ | $-1.05_{\pm0.28}$ | $-1.20_{\pm0.20}$ | $-1.80_{\pm0.31}$ |
| | $\text{PIS}_\theta + \text{VarGrad}_\varphi$ (learnt var.) | $-4.83_{\pm0.00}$ | $-4.83_{\pm0.00}$ | $-2.25_{\pm0.25}$ | $-2.04_{\pm0.01}$ | $-2.13_{\pm0.47}$ | $-2.16_{\pm0.54}$ |

| | | EUBO (↓) | | | | | |
|---|---|---|---|---|---|---|---|
| Energy ↓ | Method ↓ Steps → | $T = 2$ | $T = 3$ | $T = 4$ | $T = 5$ | $T = 10$ | $T = 20$ |
| | $\text{TB}_\theta$ (fixed var.) | $10.71_{\pm0.33}$ | $9.51_{\pm0.08}$ | $8.82_{\pm0.14}$ | $8.15_{\pm0.15}$ | $6.04_{\pm0.32}$ | $4.86_{\pm0.88}$ |
| | $\text{TB}_\theta + \text{TLM}_\varphi$ (fixed var.) | $13.00_{\pm0.55}$ | $6.08_{\pm0.12}$ | $4.46_{\pm0.21}$ | $3.03_{\pm0.09}$ | $1.26_{\pm0.16}$ | $0.63_{\pm0.02}$ |
| | $\text{TB}_{\theta,\varphi}$ (fixed var.) | $14.23_{\pm0.46}$ | $10.75_{\pm0.27}$ | $6.18_{\pm0.13}$ | $4.89_{\pm0.22}$ | $1.27_{\pm0.08}$ | $0.69_{\pm0.08}$ |
| | $\text{TB}_\theta$ (learnt var.) | $2.19_{\pm0.07}$ | $1.15_{\pm0.04}$ | $0.77_{\pm0.03}$ | $0.56_{\pm0.02}$ | $0.27_{\pm0.01}$ | $0.18_{\pm0.03}$ |
| 125GMM | $\text{TB}_\theta + \text{TLM}_\varphi$ (learnt var.) | $1.58_{\pm0.36}$ | $1.01_{\pm0.02}$ | $0.81_{\pm0.03}$ | $0.52_{\pm0.06}$ | $0.30_{\pm0.06}$ | $0.50_{\pm0.53}$ |
| | $\text{TB}_{\theta,\varphi}$ (learnt var.) | $2.06_{\pm0.31}$ | $1.15_{\pm0.10}$ | $0.72_{\pm0.03}$ | $0.51_{\pm0.04}$ | $0.21_{\pm0.05}$ | $0.06_{\pm0.00}$ |
| | $\text{PIS}_\theta$ (fixed var.) | $11.68_{\pm0.69}$ | $10.71_{\pm0.70}$ | $11.68_{\pm0.61}$ | $11.68_{\pm0.61}$ | $12.47_{\pm0.44}$ | $12.83_{\pm0.20}$ |
| | $\text{PIS}_\theta$ (learnt var.) | $11.70_{\pm1.57}$ | $12.70_{\pm0.23}$ | $12.81_{\pm0.15}$ | $12.70_{\pm0.15}$ | $12.57_{\pm0.20}$ | $11.61_{\pm0.57}$ |
| | $\text{PIS}_\theta + \text{TLM}_\varphi$ (learnt var.) | $4.98_{\pm0.86}$ | $4.53_{\pm0.36}$ | $4.01_{\pm0.39}$ | $3.30_{\pm1.56}$ | $3.67_{\pm0.87}$ | $4.37_{\pm0.44}$ |
| | $\text{PIS}_\theta + \text{VarGrad}_\varphi$ (learnt var.) | $8.80_{\pm2.96}$ | $10.50_{\pm3.13}$ | $11.89_{\pm1.12}$ | $11.42_{\pm1.41}$ | $7.88_{\pm3.26}$ | $7.60_{\pm3.75}$ |

| | | 2-Wasserstein (↓) | | | | | |
|---|---|---|---|---|---|---|---|
| Energy ↓ | Method ↓ Steps → | $T = 2$ | $T = 3$ | $T = 4$ | $T = 5$ | $T = 10$ | $T = 20$ |
| | $\text{TB}_\theta$ (fixed var.) | $8.12_{\pm0.12}$ | $7.61_{\pm0.15}$ | $7.26_{\pm0.14}$ | $6.95_{\pm0.10}$ | $6.28_{\pm0.03}$ | $5.65_{\pm0.23}$ |
| | $\text{TB}_\theta + \text{TLM}_\varphi$ (fixed var.) | $7.76_{\pm0.35}$ | $6.49_{\pm0.15}$ | $5.85_{\pm0.04}$ | $5.25_{\pm0.13}$ | $3.48_{\pm0.12}$ | $2.49_{\pm0.06}$ |
| | $\text{TB}_{\theta,\varphi}$ (fixed var.) | $7.68_{\pm0.16}$ | $6.80_{\pm0.18}$ | $6.01_{\pm0.11}$ | $5.67_{\pm0.17}$ | $3.52_{\pm0.09}$ | $2.63_{\pm0.12}$ |
| | $\text{TB}_\theta$ (learnt var.) | $3.46_{\pm0.09}$ | $3.09_{\pm0.13}$ | $2.95_{\pm0.20}$ | $2.74_{\pm0.04}$ | $2.43_{\pm0.05}$ | $1.93_{\pm0.05}$ |
| | $\text{TB}_\theta + \text{TLM}_\varphi$ (learnt var.) | $1.97_{\pm0.16}$ | $2.82_{\pm0.15}$ | $2.60_{\pm0.14}$ | $2.23_{\pm0.31}$ | $2.27_{\pm0.36}$ | $2.37_{\pm1.04}$ |
| 125GMM | $\text{TB}_{\theta,\varphi}$ (learnt var.) | $2.54_{\pm0.46}$ | $2.95_{\pm0.29}$ | $2.47_{\pm0.08}$ | $2.21_{\pm0.19}$ | $1.85_{\pm0.11}$ | $1.84_{\pm0.09}$ |
| | $\text{PIS}_\theta$ (fixed var.) | $7.60_{\pm0.14}$ | $7.01_{\pm0.36}$ | $7.32_{\pm0.54}$ | $6.91_{\pm0.48}$ | $7.19_{\pm0.31}$ | $7.94_{\pm0.47}$ |
| | $\text{PIS}_\theta$ (learnt var.) | $7.28_{\pm1.92}$ | $7.32_{\pm0.59}$ | $5.82_{\pm0.06}$ | $5.91_{\pm0.14}$ | $5.73_{\pm0.07}$ | $6.12_{\pm0.26}$ |
| | $\text{PIS}_\theta + \text{TLM}_\varphi$ (learnt var.) | $3.25_{\pm0.82}$ | $4.61_{\pm0.33}$ | $4.96_{\pm0.05}$ | $4.24_{\pm0.85}$ | $5.01_{\pm0.18}$ | $5.34_{\pm0.20}$ |
| | $\text{PIS}_\theta + \text{VarGrad}_\varphi$ (learnt var.) | $6.97_{\pm1.23}$ | $7.83_{\pm1.06}$ | $5.68_{\pm0.07}$ | $5.54_{\pm0.04}$ | $6.22_{\pm1.09}$ | $6.38_{\pm1.20}$ |
| | Ground Truth | $1.81_{\pm0.11}$ | | | | | |

Table 5: Comparison of 4 algorithms by ELBO, EUBO and 2-Wasserstein between generated and ground truth samples with varying number of discretisation steps on 25GMM. Mean and std over 3 runs are specified.

| | | | $T=2$ | $T=3$ | $T=4$ | $T=5$ | $T=10$ | $T=20$ |
|---|---|---|---|---|---|---|---|---|
| | | | ELBO ($\uparrow$) | | | | | |
| Energy $\downarrow$ | Method $\downarrow$ Steps $\rightarrow$ | | $T=2$ | $T=3$ | $T=4$ | $T=5$ | $T=10$ | $T=20$ |
| | $\text{TB}_\theta$ | (fixed var.) | $-4.92_{\pm0.06}$ | $-3.53_{\pm0.05}$ | $-2.80_{\pm0.03}$ | $-2.35_{\pm0.04}$ | $-1.53_{\pm0.01}$ | $-1.11_{\pm0.02}$ |
| | $\text{TB}_\theta + \text{TLM}_\varphi$ | (fixed var.) | $-4.75_{\pm0.05}$ | $-3.31_{\pm0.05}$ | $-2.39_{\pm0.02}$ | $-1.79_{\pm0.05}$ | $-0.70_{\pm0.04}$ | $-0.37_{\pm0.03}$ |
| | $\text{TB}_{\theta,\varphi}$ | (fixed var.) | $-5.34_{\pm0.11}$ | $-3.76_{\pm0.06}$ | $-2.61_{\pm0.04}$ | $-2.01_{\pm0.06}$ | $-0.75_{\pm0.05}$ | $-0.37_{\pm0.03}$ |
| | $\text{TB}_\theta$ | (learnt var.) | $-3.10_{\pm0.07}$ | $-1.44_{\pm0.07}$ | $-0.80_{\pm0.07}$ | $-0.54_{\pm0.02}$ | $-0.30_{\pm0.15}$ | $-0.12_{\pm0.02}$ |
| | $\text{TB}_\theta + \text{TLM}_\varphi$ | (learnt var.) | $-2.05_{\pm0.55}$ | $-1.01_{\pm0.09}$ | $-0.59_{\pm0.01}$ | $-0.36_{\pm0.01}$ | $-0.07_{\pm0.01}$ | $-0.18_{\pm0.10}$ |
| 25GMM | $\text{TB}_{\theta,\varphi}$ | (learnt var.) | $-1.83_{\pm0.04}$ | $-1.19_{\pm0.03}$ | $-0.70_{\pm0.01}$ | $-0.42_{\pm0.03}$ | $-0.09_{\pm0.02}$ | $-0.03_{\pm0.01}$ |
| | $\text{PIS}_\theta$ | (fixed var.) | $-4.36_{\pm0.04}$ | $-3.23_{\pm0.01}$ | $-2.65_{\pm0.01}$ | $-2.36_{\pm0.04}$ | $-1.92_{\pm0.09}$ | $-1.64_{\pm0.14}$ |
| | $\text{PIS}_\theta$ | (learnt var.) | $-2.14_{\pm0.81}$ | $-1.45_{\pm0.02}$ | $-1.37_{\pm0.01}$ | $-1.31_{\pm0.01}$ | $-1.20_{\pm0.01}$ | $-1.34_{\pm0.13}$ |
| | $\text{PIS}_\theta + \text{TLM}_\varphi$ | (learnt var.) | $-1.13_{\pm0.19}$ | $-0.76_{\pm0.02}$ | $-0.82_{\pm0.16}$ | $-1.13_{\pm0.01}$ | $-1.01_{\pm0.17}$ | $-1.81_{\pm0.30}$ |
| | $\text{PIS}_\theta + \text{VarGrad}_\varphi$ | (learnt var.) | $-2.37_{\pm0.00}$ | $-1.75_{\pm0.38}$ | $-1.29_{\pm0.01}$ | $-1.18_{\pm0.10}$ | $-1.19_{\pm0.01}$ | $-1.13_{\pm0.01}$ |
| | | | EUBO ($\downarrow$) | | | | | |
| Energy $\downarrow$ | Method $\downarrow$ Steps $\rightarrow$ | | $T=2$ | $T=3$ | $T=4$ | $T=5$ | $T=10$ | $T=20$ |
| | $\text{TB}_\theta$ | (fixed var.) | $7.01_{\pm0.08}$ | $5.91_{\pm0.07}$ | $5.33_{\pm0.13}$ | $5.07_{\pm0.22}$ | $4.11_{\pm0.52}$ | $2.43_{\pm0.13}$ |
| | $\text{TB}_\theta + \text{TLM}_\varphi$ | (fixed var.) | $7.25_{\pm0.50}$ | $3.57_{\pm0.08}$ | $2.33_{\pm0.07}$ | $1.64_{\pm0.06}$ | $0.67_{\pm0.02}$ | $0.35_{\pm0.01}$ |
| | $\text{TB}_{\theta,\varphi}$ | (fixed var.) | $9.39_{\pm0.19}$ | $7.77_{\pm0.96}$ | $3.93_{\pm0.66}$ | $2.67_{\pm0.45}$ | $0.70_{\pm0.06}$ | $0.35_{\pm0.01}$ |
| | $\text{TB}_\theta$ | (learnt var.) | $1.41_{\pm0.04}$ | $0.74_{\pm0.01}$ | $0.47_{\pm0.01}$ | $0.35_{\pm0.01}$ | $0.90_{\pm1.05}$ | $0.09_{\pm0.00}$ |
| | $\text{TB}_\theta + \text{TLM}_\varphi$ | (learnt var.) | $0.99_{\pm0.14}$ | $0.56_{\pm0.05}$ | $0.36_{\pm0.01}$ | $0.24_{\pm0.01}$ | $0.07_{\pm0.00}$ | $1.87_{\pm1.66}$ |
| 25GMM | $\text{TB}_{\theta,\varphi}$ | (learnt var.) | $0.94_{\pm0.03}$ | $0.60_{\pm0.02}$ | $0.37_{\pm0.02}$ | $0.24_{\pm0.00}$ | $0.07_{\pm0.01}$ | $0.03_{\pm0.00}$ |
| | $\text{PIS}_\theta$ | (fixed var.) | $7.07_{\pm0.23}$ | $7.19_{\pm0.22}$ | $7.31_{\pm0.54}$ | $7.12_{\pm0.20}$ | $8.51_{\pm0.06}$ | $8.51_{\pm0.06}$ |
| | $\text{PIS}_\theta$ | (learnt var.) | $8.41_{\pm0.20}$ | $8.40_{\pm0.06}$ | $8.42_{\pm0.04}$ | $8.44_{\pm0.03}$ | $8.40_{\pm0.16}$ | $7.02_{\pm0.05}$ |
| | $\text{PIS}_\theta + \text{TLM}_\varphi$ | (learnt var.) | $6.03_{\pm0.61}$ | $5.41_{\pm0.43}$ | $3.45_{\pm0.63}$ | $5.50_{\pm2.13}$ | $4.25_{\pm1.77}$ | $7.61_{\pm1.29}$ |
| | $\text{PIS}_\theta + \text{VarGrad}_\varphi$ | (learnt var.) | $8.40_{\pm0.06}$ | $8.42_{\pm0.08}$ | $8.43_{\pm0.08}$ | $6.76_{\pm1.25}$ | $3.58_{\pm0.49}$ | $5.71_{\pm2.01}$ |
| | | | 2-Wasserstein ($\downarrow$) | | | | | |
| Energy $\downarrow$ | Method $\downarrow$ Steps $\rightarrow$ | | $T=2$ | $T=3$ | $T=4$ | $T=5$ | $T=10$ | $T=20$ |
| | $\text{TB}_\theta$ | (fixed var.) | $6.47_{\pm0.05}$ | $6.08_{\pm0.07}$ | $5.70_{\pm0.02}$ | $5.52_{\pm0.08}$ | $5.03_{\pm0.17}$ | $4.39_{\pm0.01}$ |
| | $\text{TB}_\theta + \text{TLM}_\varphi$ | (fixed var.) | $6.27_{\pm0.09}$ | $5.11_{\pm0.10}$ | $4.44_{\pm0.06}$ | $3.87_{\pm0.04}$ | $2.38_{\pm0.11}$ | $1.66_{\pm0.07}$ |
| | $\text{TB}_{\theta,\varphi}$ | (fixed var.) | $6.36_{\pm0.07}$ | $5.74_{\pm0.18}$ | $4.75_{\pm0.10}$ | $4.32_{\pm0.06}$ | $2.56_{\pm0.15}$ | $1.77_{\pm0.08}$ |
| | $\text{TB}_\theta$ | (learnt var.) | $2.47_{\pm0.03}$ | $2.31_{\pm0.10}$ | $2.18_{\pm0.06}$ | $2.04_{\pm0.10}$ | $2.11_{\pm0.90}$ | $1.46_{\pm0.30}$ |
| | $\text{TB}_\theta + \text{TLM}_\varphi$ | (learnt var.) | $1.37_{\pm0.16}$ | $1.45_{\pm0.03}$ | $1.07_{\pm0.11}$ | $1.01_{\pm0.09}$ | $0.91_{\pm0.03}$ | $2.14_{\pm1.68}$ |
| 25GMM | $\text{TB}_{\theta,\varphi}$ | (learnt var.) | $1.77_{\pm0.17}$ | $1.76_{\pm0.27}$ | $1.07_{\pm0.15}$ | $1.13_{\pm0.11}$ | $1.05_{\pm0.20}$ | $1.17_{\pm0.24}$ |
| | $\text{PIS}_\theta$ | (fixed var.) | $5.71_{\pm0.10}$ | $5.43_{\pm0.07}$ | $5.22_{\pm0.04}$ | $5.24_{\pm0.08}$ | $5.63_{\pm0.29}$ | $5.31_{\pm0.36}$ |
| | $\text{PIS}_\theta$ | (learnt var.) | $5.90_{\pm1.46}$ | $4.68_{\pm0.08}$ | $4.56_{\pm0.09}$ | $4.46_{\pm0.06}$ | $4.58_{\pm0.08}$ | $4.87_{\pm0.26}$ |
| | $\text{PIS}_\theta + \text{TLM}_\varphi$ | (learnt var.) | $4.09_{\pm0.53}$ | $3.47_{\pm0.26}$ | $3.75_{\pm0.38}$ | $4.39_{\pm0.04}$ | $4.37_{\pm0.11}$ | $6.10_{\pm1.24}$ |
| | $\text{PIS}_\theta + \text{VarGrad}_\varphi$ | (learnt var.) | $6.98_{\pm0.03}$ | $5.80_{\pm1.08}$ | $4.35_{\pm0.09}$ | $4.17_{\pm0.27}$ | $4.38_{\pm0.05}$ | $4.40_{\pm0.11}$ |
| | Ground Truth | | $1.10_{\pm0.14}$ | | | | | |

Table 6: Comparison of 4 algorithms by ELBO, EUBO and 2-Wasserstein between generated and ground truth samples with varying number of discretisation steps on Slightly Distorted 25GMM. Mean and std over 3 runs are specified.

| | | ELBO ($\uparrow$) | | | | | | |
|---|---|---|---|---|---|---|---|---|
| Energy $\downarrow$ | Method $\downarrow$ Steps $\rightarrow$ | | $T=2$ | $T=3$ | $T=4$ | $T=5$ | $T=10$ | $T=20$ |
| Slightly Distorted 25GMM | $TB_\theta$ | (fixed var.) | $-4.80_{\pm0.07}$ | $-3.54_{\pm0.09}$ | $-2.88_{\pm0.03}$ | $-2.44_{\pm0.04}$ | $-1.56_{\pm0.02}$ | $-1.12_{\pm0.02}$ |
| | $TB_\theta + TLM_\varphi$ | (fixed var.) | $-4.85_{\pm0.14}$ | $-3.44_{\pm0.02}$ | $-2.43_{\pm0.12}$ | $-1.80_{\pm0.05}$ | $-0.73_{\pm0.02}$ | $-0.37_{\pm0.02}$ |
| | $TB_{\theta,\varphi}$ | (fixed var.) | $-5.31_{\pm0.04}$ | $-3.67_{\pm0.07}$ | $-2.63_{\pm0.04}$ | $-2.03_{\pm0.06}$ | $-0.77_{\pm0.03}$ | $-0.35_{\pm0.02}$ |
| | $TB_\theta$ | (learnt var.) | $-3.12_{\pm0.13}$ | $-1.47_{\pm0.03}$ | $-0.82_{\pm0.04}$ | $-0.52_{\pm0.03}$ | $-0.22_{\pm0.01}$ | $-1.14_{\pm1.45}$ |
| | $TB_\theta + TLM_\varphi$ | (learnt var.) | $-2.16_{\pm0.57}$ | $-0.99_{\pm0.03}$ | $-0.63_{\pm0.08}$ | $-0.35_{\pm0.02}$ | $-0.12_{\pm0.04}$ | $-0.04_{\pm0.01}$ |
| | $TB_{\theta,\varphi}$ | (learnt var.) | $-2.49_{\pm0.34}$ | $-1.17_{\pm0.04}$ | $-0.66_{\pm0.05}$ | $-0.42_{\pm0.03}$ | $-0.11_{\pm0.02}$ | $-0.04_{\pm0.00}$ |
| | $PIS_\theta$ | (fixed var.) | $-4.15_{\pm0.05}$ | $-3.39_{\pm0.04}$ | $-3.14_{\pm0.45}$ | $-2.51_{\pm0.09}$ | $-1.97_{\pm0.09}$ | $-1.80_{\pm0.07}$ |
| | $PIS_\theta$ | (learnt var.) | $-1.21_{\pm0.27}$ | $-1.47_{\pm0.02}$ | $-1.40_{\pm0.01}$ | $-1.31_{\pm0.02}$ | $-1.29_{\pm0.05}$ | $-1.51_{\pm0.19}$ |
| | $PIS_\theta + TLM_\varphi$ | (learnt var.) | $-0.97_{\pm0.21}$ | $-0.62_{\pm0.19}$ | $-0.61_{\pm0.09}$ | $-0.92_{\pm0.17}$ | $-1.08_{\pm0.09}$ | $-1.25_{\pm0.10}$ |
| | $PIS_\theta + VarGrad_\varphi$ | (learnt var.) | $-2.38_{\pm0.69}$ | $-1.48_{\pm0.15}$ | $-1.30_{\pm0.01}$ | $-1.22_{\pm0.10}$ | $-1.04_{\pm0.11}$ | $-1.19_{\pm0.05}$ |

| | | EUBO ($\downarrow$) | | | | | | |
|---|---|---|---|---|---|---|---|---|
| Energy $\downarrow$ | Method $\downarrow$ Steps $\rightarrow$ | | $T=2$ | $T=3$ | $T=4$ | $T=5$ | $T=10$ | $T=20$ |
| Slightly Distorted 25GMM | $TB_\theta$ | (fixed var.) | $7.41_{\pm0.17}$ | $6.20_{\pm0.09}$ | $5.38_{\pm0.17}$ | $4.69_{\pm0.15}$ | $3.71_{\pm0.11}$ | $2.60_{\pm0.12}$ |
| | $TB_\theta + TLM_\varphi$ | (fixed var.) | $7.27_{\pm0.26}$ | $3.58_{\pm0.04}$ | $2.28_{\pm0.03}$ | $1.62_{\pm0.07}$ | $0.68_{\pm0.01}$ | $0.35_{\pm0.00}$ |
| | $TB_{\theta,\varphi}$ | (fixed var.) | $9.50_{\pm0.69}$ | $6.46_{\pm0.55}$ | $3.70_{\pm0.67}$ | $2.47_{\pm0.40}$ | $0.69_{\pm0.01}$ | $0.35_{\pm0.01}$ |
| | $TB_\theta$ | (learnt var.) | $1.39_{\pm0.03}$ | $0.74_{\pm0.01}$ | $0.47_{\pm0.01}$ | $0.35_{\pm0.01}$ | $0.16_{\pm0.00}$ | $7.06_{\pm9.86}$ |
| | $TB_\theta + TLM_\varphi$ | (learnt var.) | $1.06_{\pm0.32}$ | $0.56_{\pm0.03}$ | $0.40_{\pm0.01}$ | $0.26_{\pm0.02}$ | $0.11_{\pm0.03}$ | $0.04_{\pm0.01}$ |
| | $TB_{\theta,\varphi}$ | (learnt var.) | $1.13_{\pm0.11}$ | $0.57_{\pm0.02}$ | $0.38_{\pm0.02}$ | $0.26_{\pm0.02}$ | $0.09_{\pm0.01}$ | $0.03_{\pm0.00}$ |
| | $PIS_\theta$ | (fixed var.) | $7.58_{\pm0.23}$ | $8.16_{\pm0.14}$ | $8.33_{\pm0.21}$ | $8.38_{\pm0.08}$ | $8.04_{\pm0.58}$ | $8.37_{\pm0.13}$ |
| | $PIS_\theta$ | (learnt var.) | $8.05_{\pm0.54}$ | $8.38_{\pm0.05}$ | $8.38_{\pm0.05}$ | $8.41_{\pm0.03}$ | $7.79_{\pm0.46}$ | $7.52_{\pm0.22}$ |
| | $PIS_\theta + TLM_\varphi$ | (learnt var.) | $5.81_{\pm1.88}$ | $2.37_{\pm0.42}$ | $3.90_{\pm0.12}$ | $3.59_{\pm1.95}$ | $2.96_{\pm0.33}$ | $6.01_{\pm1.02}$ |
| | $PIS_\theta + VarGrad_\varphi$ | (learnt var.) | $8.37_{\pm0.05}$ | $8.37_{\pm0.05}$ | $7.69_{\pm1.00}$ | $5.10_{\pm1.33}$ | $3.26_{\pm0.14}$ | $3.79_{\pm0.93}$ |

| | | 2-Wasserstein ($\downarrow$) | | | | | | |
|---|---|---|---|---|---|---|---|---|
| Energy $\downarrow$ | Method $\downarrow$ Steps $\rightarrow$ | | $T=2$ | $T=3$ | $T=4$ | $T=5$ | $T=10$ | $T=20$ |
| Slightly Distorted 25GMM | $TB_\theta$ | (fixed var.) | $6.78_{\pm0.05}$ | $6.40_{\pm0.03}$ | $5.82_{\pm0.11}$ | $5.59_{\pm0.02}$ | $5.16_{\pm0.13}$ | $4.66_{\pm0.09}$ |
| | $TB_\theta + TLM_\varphi$ | (fixed var.) | $6.33_{\pm0.10}$ | $5.23_{\pm0.09}$ | $4.46_{\pm0.05}$ | $3.81_{\pm0.05}$ | $2.35_{\pm0.04}$ | $1.64_{\pm0.20}$ |
| | $TB_{\theta,\varphi}$ | (fixed var.) | $6.28_{\pm0.10}$ | $5.53_{\pm0.09}$ | $4.86_{\pm0.26}$ | $4.28_{\pm0.17}$ | $2.51_{\pm0.04}$ | $1.57_{\pm0.01}$ |
| | $TB_\theta$ | (learnt var.) | $2.55_{\pm0.06}$ | $2.27_{\pm0.06}$ | $2.17_{\pm0.10}$ | $1.92_{\pm0.12}$ | $1.62_{\pm0.12}$ | $6.66_{\pm7.38}$ |
| | $TB_\theta + TLM_\varphi$ | (learnt var.) | $1.56_{\pm0.17}$ | $1.54_{\pm0.22}$ | $1.14_{\pm0.15}$ | $0.98_{\pm0.10}$ | $1.15_{\pm0.19}$ | $0.83_{\pm0.06}$ |
| | $TB_{\theta,\varphi}$ | (learnt var.) | $2.06_{\pm0.27}$ | $1.52_{\pm0.11}$ | $1.28_{\pm0.20}$ | $1.04_{\pm0.13}$ | $1.10_{\pm0.20}$ | $1.04_{\pm0.13}$ |
| | $PIS_\theta$ | (fixed var.) | $6.40_{\pm0.39}$ | $5.83_{\pm0.14}$ | $5.88_{\pm0.46}$ | $5.73_{\pm0.25}$ | $5.47_{\pm0.42}$ | $6.14_{\pm0.37}$ |
| | $PIS_\theta$ | (learnt var.) | $4.36_{\pm0.92}$ | $4.76_{\pm0.03}$ | $4.61_{\pm0.07}$ | $4.56_{\pm0.14}$ | $4.63_{\pm0.11}$ | $4.94_{\pm0.27}$ |
| | $PIS_\theta + TLM_\varphi$ | (learnt var.) | $3.56_{\pm0.64}$ | $3.15_{\pm0.52}$ | $3.22_{\pm0.14}$ | $3.98_{\pm0.42}$ | $4.37_{\pm0.19}$ | $4.64_{\pm0.16}$ |
| | $PIS_\theta + VarGrad_\varphi$ | (learnt var.) | $6.17_{\pm1.16}$ | $4.92_{\pm0.75}$ | $4.33_{\pm0.05}$ | $4.31_{\pm0.19}$ | $4.24_{\pm0.12}$ | $4.45_{\pm0.09}$ |
| | Ground Truth | | | | $1.06_{\pm0.14}$ | | | |

Table 7: Comparison of 4 algorithms by ELBO, EUBO and 2-Wasserstein between generated and ground truth samples with varying number of discretisation steps on Distorted 25GMM. Mean and std over 3 runs are specified.

| ELBO ($\uparrow$) | | | | | | | | |
|---|---|---|---|---|---|---|---|---|
| Energy $\downarrow$ | Method $\downarrow$ Steps $\rightarrow$ | | $T = 2$ | $T = 3$ | $T = 4$ | $T = 5$ | $T = 10$ | $T = 20$ |
| | $TB_\theta$ | (fixed var.) | $-4.48_{\pm0.07}$ | $-3.51_{\pm0.05}$ | $-2.88_{\pm0.03}$ | $-2.45_{\pm0.02}$ | $-1.60_{\pm0.03}$ | $-1.13_{\pm0.01}$ |
| | $TB_\theta + TLM_\varphi$ | (fixed var.) | $-4.58_{\pm0.06}$ | $-3.48_{\pm0.04}$ | $-2.45_{\pm0.09}$ | $-1.87_{\pm0.05}$ | $-0.77_{\pm0.01}$ | $-0.37_{\pm0.00}$ |
| | $TB_{\theta,\varphi}$ | (fixed var.) | $-5.47_{\pm0.07}$ | $-3.90_{\pm0.08}$ | $-2.77_{\pm0.02}$ | $-2.10_{\pm0.06}$ | $-0.81_{\pm0.02}$ | $-0.39_{\pm0.03}$ |
| | $TB_\theta$ | (learnt var.) | $-3.09_{\pm0.02}$ | $-1.47_{\pm0.05}$ | $-0.84_{\pm0.05}$ | $-0.57_{\pm0.02}$ | $-0.22_{\pm0.02}$ | $-0.12_{\pm0.02}$ |
| | $TB_\theta + TLM_\varphi$ | (learnt var.) | $-2.20_{\pm0.31}$ | $-1.18_{\pm0.05}$ | $-0.63_{\pm0.04}$ | $-0.34_{\pm0.02}$ | $-0.12_{\pm0.02}$ | $-0.04_{\pm0.01}$ |
| Distorted | $TB_{\theta,\varphi}$ | (learnt var.) | $-2.27_{\pm0.08}$ | $-1.37_{\pm0.04}$ | $-0.77_{\pm0.06}$ | $-0.46_{\pm0.01}$ | $-0.12_{\pm0.01}$ | $-0.05_{\pm0.00}$ |
| 25GMM | $PIS_\theta$ | (fixed var.) | $-4.04_{\pm0.17}$ | $-3.28_{\pm0.08}$ | $-2.79_{\pm0.13}$ | $-2.60_{\pm0.03}$ | $-2.09_{\pm0.04}$ | $-2.09_{\pm0.26}$ |
| | $PIS_\theta$ | (learnt var.) | $-1.33_{\pm0.17}$ | $-1.71_{\pm0.16}$ | $-1.52_{\pm0.15}$ | $-1.78_{\pm0.16}$ | $-1.65_{\pm0.23}$ | $-1.91_{\pm0.06}$ |
| | $PIS_\theta + TLM_\varphi$ | (learnt var.) | $-1.32_{\pm0.23}$ | $-1.25_{\pm0.47}$ | $-1.06_{\pm0.11}$ | $-1.05_{\pm0.34}$ | $-1.31_{\pm0.24}$ | $-1.62_{\pm0.13}$ |
| | $PIS_\theta + VarGrad_\varphi$ | (learnt var.) | $-1.89_{\pm0.36}$ | $-1.32_{\pm0.55}$ | $-1.41_{\pm0.20}$ | $-1.37_{\pm0.21}$ | $-1.29_{\pm0.03}$ | $-1.43_{\pm0.09}$ |

| EUBO ($\downarrow$) | | | | | | | | |
|---|---|---|---|---|---|---|---|---|
| Energy $\downarrow$ | Method $\downarrow$ Steps $\rightarrow$ | | $T = 2$ | $T = 3$ | $T = 4$ | $T = 5$ | $T = 10$ | $T = 20$ |
| | $TB_\theta$ | (fixed var.) | $9.00_{\pm0.41}$ | $7.08_{\pm0.41}$ | $5.93_{\pm0.32}$ | $5.23_{\pm0.17}$ | $3.85_{\pm0.35}$ | $2.80_{\pm0.17}$ |
| | $TB_\theta + TLM_\varphi$ | (fixed var.) | $7.23_{\pm0.45}$ | $3.72_{\pm0.22}$ | $2.43_{\pm0.08}$ | $1.77_{\pm0.07}$ | $0.71_{\pm0.03}$ | $0.36_{\pm0.01}$ |
| | $TB_{\theta,\varphi}$ | (fixed var.) | $9.47_{\pm0.50}$ | $6.94_{\pm1.20}$ | $4.04_{\pm0.20}$ | $2.54_{\pm0.20}$ | $0.77_{\pm0.05}$ | $0.36_{\pm0.02}$ |
| | $TB_\theta$ | (learnt var.) | $1.44_{\pm0.03}$ | $0.78_{\pm0.02}$ | $0.49_{\pm0.00}$ | $0.37_{\pm0.01}$ | $0.17_{\pm0.01}$ | $0.10_{\pm0.00}$ |
| Distorted | $TB_\theta + TLM_\varphi$ | (learnt var.) | $1.13_{\pm0.15}$ | $0.73_{\pm0.04}$ | $0.41_{\pm0.02}$ | $0.27_{\pm0.01}$ | $0.11_{\pm0.01}$ | $0.04_{\pm0.00}$ |
| 25GMM | $TB_{\theta,\varphi}$ | (learnt var.) | $1.16_{\pm0.03}$ | $0.79_{\pm0.03}$ | $0.46_{\pm0.02}$ | $0.29_{\pm0.01}$ | $0.09_{\pm0.00}$ | $0.05_{\pm0.00}$ |
| | $PIS_\theta$ | (fixed var.) | $8.47_{\pm0.05}$ | $8.26_{\pm0.17}$ | $8.47_{\pm0.05}$ | $8.20_{\pm0.43}$ | $8.38_{\pm0.09}$ | $8.47_{\pm0.05}$ |
| | $PIS_\theta$ | (learnt var.) | $8.40_{\pm0.10}$ | $8.36_{\pm0.05}$ | $8.36_{\pm0.05}$ | $8.39_{\pm0.03}$ | $8.30_{\pm0.06}$ | $8.43_{\pm0.07}$ |
| | $PIS_\theta + TLM_\varphi$ | (learnt var.) | $8.36_{\pm0.05}$ | $8.22_{\pm0.24}$ | $5.70_{\pm0.78}$ | $7.14_{\pm0.91}$ | $5.13_{\pm2.35}$ | $8.39_{\pm0.08}$ |
| | $PIS_\theta + VarGrad_\varphi$ | (learnt var.) | $8.36_{\pm0.05}$ | $7.68_{\pm0.99}$ | $8.10_{\pm0.37}$ | $6.13_{\pm1.57}$ | $5.09_{\pm0.97}$ | $7.57_{\pm0.69}$ |

| 2-Wasserstein ($\downarrow$) | | | | | | | | |
|---|---|---|---|---|---|---|---|---|
| Energy $\downarrow$ | Method $\downarrow$ Steps $\rightarrow$ | | $T = 2$ | $T = 3$ | $T = 4$ | $T = 5$ | $T = 10$ | $T = 20$ |
| | $TB_\theta$ | (fixed var.) | $8.29_{\pm0.30}$ | $7.35_{\pm0.12}$ | $6.52_{\pm0.12}$ | $6.18_{\pm0.11}$ | $5.40_{\pm0.30}$ | $4.84_{\pm0.10}$ |
| | $TB_\theta + TLM_\varphi$ | (fixed var.) | $7.24_{\pm0.38}$ | $5.69_{\pm0.17}$ | $4.69_{\pm0.26}$ | $3.89_{\pm0.05}$ | $2.45_{\pm0.05}$ | $1.67_{\pm0.03}$ |
| | $TB_{\theta,\varphi}$ | (fixed var.) | $6.70_{\pm0.20}$ | $5.95_{\pm0.09}$ | $4.95_{\pm0.14}$ | $4.32_{\pm0.03}$ | $2.64_{\pm0.05}$ | $1.69_{\pm0.12}$ |
| | $TB_\theta$ | (learnt var.) | $2.68_{\pm0.06}$ | $2.28_{\pm0.08}$ | $2.16_{\pm0.12}$ | $2.05_{\pm0.04}$ | $1.74_{\pm0.05}$ | $1.33_{\pm0.10}$ |
| Distorted | $TB_\theta + TLM_\varphi$ | (learnt var.) | $2.03_{\pm0.23}$ | $1.58_{\pm0.12}$ | $1.25_{\pm0.12}$ | $1.00_{\pm0.20}$ | $0.97_{\pm0.11}$ | $0.92_{\pm0.11}$ |
| 25GMM | $TB_{\theta,\varphi}$ | (learnt var.) | $2.37_{\pm0.17}$ | $1.78_{\pm0.14}$ | $1.40_{\pm0.21}$ | $1.11_{\pm0.21}$ | $1.21_{\pm0.26}$ | $1.01_{\pm0.12}$ |
| | $PIS_\theta$ | (fixed var.) | $6.99_{\pm0.05}$ | $6.07_{\pm0.35}$ | $6.11_{\pm0.49}$ | $5.86_{\pm0.11}$ | $5.90_{\pm0.09}$ | $6.68_{\pm0.45}$ |
| | $PIS_\theta$ | (learnt var.) | $4.55_{\pm0.78}$ | $5.60_{\pm0.74}$ | $5.14_{\pm0.73}$ | $6.38_{\pm0.43}$ | $5.57_{\pm0.58}$ | $6.61_{\pm0.27}$ |
| | $PIS_\theta + TLM_\varphi$ | (learnt var.) | $5.14_{\pm1.30}$ | $4.99_{\pm1.40}$ | $4.12_{\pm0.19}$ | $4.43_{\pm1.15}$ | $5.14_{\pm0.73}$ | $5.47_{\pm0.66}$ |
| | $PIS_\theta + VarGrad_\varphi$ | (learnt var.) | $5.98_{\pm1.38}$ | $4.43_{\pm1.86}$ | $4.87_{\pm0.92}$ | $4.84_{\pm0.83}$ | $4.65_{\pm0.06}$ | $4.95_{\pm0.36}$ |
| | Ground Truth | | | | $1.06_{\pm0.14}$ | | | |

Table 8: Comparison of 4 algorithms by ELBO, EUBO, and 2-Wasserstein between generated and ground truth samples with varying number of discretisation steps on 40GMM. Mean and std over 3 runs are specified.

### ELBO ($\uparrow$)

| Energy $\downarrow$ | Method $\downarrow$ Steps $\rightarrow$ | | $T=2$ | $T=3$ | $T=4$ | $T=5$ | $T=10$ | $T=15$ | $T=20$ | $T=25$ |
|---|---|---|---|---|---|---|---|---|---|---|
| 40GMM | $TB_\theta$ | (fixed var.) | $-4.52_{\pm1.06}$ | $-12.95_{\pm13.36}$ | $-3.75_{\pm0.81}$ | $-3.60_{\pm1.03}$ | $-2.40_{\pm0.06}$ | $-2.30_{\pm0.02}$ | $-2.83_{\pm0.82}$ | $-2.29_{\pm0.12}$ |
| | $TB_\theta + TLM_\varphi$ | (fixed var.) | $-4.96_{\pm0.78}$ | $-3.11_{\pm0.21}$ | $-2.62_{\pm0.09}$ | $-2.46_{\pm0.05}$ | $-2.17_{\pm0.01}$ | $-2.04_{\pm0.07}$ | $-1.98_{\pm0.04}$ | $-1.96_{\pm0.04}$ |
| | $TB_{\theta,\varphi}$ | (fixed var.) | $-4.18_{\pm1.18}$ | $-3.35_{\pm0.02}$ | $-2.81_{\pm0.07}$ | $-2.48_{\pm0.06}$ | $-2.26_{\pm0.03}$ | $-2.06_{\pm0.08}$ | $-1.95_{\pm0.09}$ | $-2.00_{\pm0.08}$ |
| | $TB_\theta$ | (learnt var.) | $-5.30_{\pm0.65}$ | $-3.96_{\pm0.92}$ | $-4.96_{\pm1.15}$ | $-3.58_{\pm0.04}$ | $-2.88_{\pm0.64}$ | $-2.26_{\pm0.13}$ | $-2.08_{\pm0.14}$ | $-1.94_{\pm0.04}$ |
| | $TB_\theta + TLM_\varphi$ | (learnt var.) | $-3.76_{\pm0.34}$ | $-2.64_{\pm0.12}$ | $-2.03_{\pm0.06}$ | $-1.74_{\pm0.03}$ | $-1.28_{\pm0.03}$ | $-1.14_{\pm0.13}$ | $-1.25_{\pm0.19}$ | $-1.34_{\pm0.15}$ |
| | $TB_{\theta,\varphi}$ | (learnt var.) | $-4.12_{\pm0.31}$ | $-2.60_{\pm0.12}$ | $-2.44_{\pm0.17}$ | $-1.90_{\pm0.03}$ | $-1.26_{\pm0.11}$ | $-1.15_{\pm0.10}$ | $-0.99_{\pm0.09}$ | $-1.03_{\pm0.21}$ |
| | $PIS_\theta$ | (fixed var.) | $-4.45_{\pm0.03}$ | $-3.67_{\pm0.70}$ | $-3.56_{\pm0.66}$ | $-3.95_{\pm0.00}$ | $-3.83_{\pm0.01}$ | $-3.33_{\pm0.64}$ | $-3.31_{\pm0.64}$ | $-3.76_{\pm0.00}$ |
| | $PIS_\theta$ | (learnt var.) | $-3.34_{\pm0.49}$ | $-2.93_{\pm0.53}$ | $-2.90_{\pm0.55}$ | $-2.87_{\pm0.58}$ | $-2.38_{\pm0.01}$ | $-3.20_{\pm0.69}$ | $-2.77_{\pm0.65}$ | $-2.29_{\pm0.08}$ |
| | $PIS_\theta + TLM_\varphi$ | (learnt var.) | $-2.53_{\pm0.11}$ | $-2.62_{\pm0.00}$ | $-2.22_{\pm0.08}$ | $-2.34_{\pm0.02}$ | $-2.29_{\pm0.05}$ | $-2.33_{\pm0.01}$ | $-2.32_{\pm0.01}$ | $-2.32_{\pm0.01}$ |
| | $PIS_\theta + VarGrad_\varphi$ | (learnt var.) | $-2.69_{\pm0.04}$ | $-2.55_{\pm0.13}$ | $-2.47_{\pm0.06}$ | $-2.79_{\pm0.64}$ | $-2.73_{\pm0.69}$ | $-2.72_{\pm0.69}$ | $-2.27_{\pm0.09}$ | $-2.26_{\pm0.09}$ |

### EUBO ($\downarrow$)

| Energy $\downarrow$ | Method $\downarrow$ Steps $\rightarrow$ | | $T=2$ | $T=3$ | $T=4$ | $T=5$ | $T=10$ | $T=15$ | $T=20$ | $T=25$ |
|---|---|---|---|---|---|---|---|---|---|---|
| 40GMM | $TB_\theta$ | (fixed var.) | $141.80_{\pm22.85}$ | $97.40_{\pm4.91}$ | $251.06_{\pm95.60}$ | $102.98_{\pm29.73}$ | $102.84_{\pm10.99}$ | $2732.36_{\pm3753.31}$ | $121.65_{\pm26.31}$ | $98.32_{\pm22.12}$ |
| | $TB_\theta + TLM_\varphi$ | (fixed var.) | $76.05_{\pm2.25}$ | $70.30_{\pm14.86}$ | $98.52_{\pm7.47}$ | $82.80_{\pm8.02}$ | $117.96_{\pm8.84}$ | $114.55_{\pm12.78}$ | $108.29_{\pm18.38}$ | $84.96_{\pm23.71}$ |
| | $TB_{\theta,\varphi}$ | (fixed var.) | $87.03_{\pm26.30}$ | $58.65_{\pm2.54}$ | $97.50_{\pm17.17}$ | $104.46_{\pm15.08}$ | $116.41_{\pm12.13}$ | $116.25_{\pm34.53}$ | $115.79_{\pm24.49}$ | $107.80_{\pm6.42}$ |
| | $TB_\theta$ | (learnt var.) | $5.33_{\pm0.15}$ | $7.86_{\pm2.26}$ | $4.44_{\pm0.77}$ | $38.06_{\pm47.29}$ | $5.98_{\pm2.29}$ | $6.89_{\pm4.76}$ | $5.29_{\pm3.08}$ | $37.75_{\pm46.22}$ |
| | $TB_\theta + TLM_\varphi$ | (learnt var.) | $2.74_{\pm0.19}$ | $3.38_{\pm0.18}$ | $3.88_{\pm0.25}$ | $6.11_{\pm1.44}$ | $7.25_{\pm0.88}$ | $5.84_{\pm0.50}$ | $8.26_{\pm2.03}$ | $14.95_{\pm10.41}$ |
| | $TB_{\theta,\varphi}$ | (learnt var.) | $3.17_{\pm0.14}$ | $3.20_{\pm0.52}$ | $3.18_{\pm0.29}$ | $5.34_{\pm0.66}$ | $4.56_{\pm0.22}$ | $4.64_{\pm0.17}$ | $4.70_{\pm0.36}$ | $5.90_{\pm1.03}$ |
| | $PIS_\theta$ | (fixed var.) | $108.22_{\pm0.38}$ | $106.93_{\pm1.45}$ | $106.20_{\pm2.48}$ | $108.22_{\pm0.38}$ | $108.22_{\pm0.38}$ | $108.22_{\pm0.38}$ | $106.84_{\pm1.57}$ | $106.39_{\pm2.21}$ |
| | $PIS_\theta$ | (learnt var.) | $106.96_{\pm0.43}$ | $106.97_{\pm0.45}$ | $104.85_{\pm2.55}$ | $106.97_{\pm0.46}$ | $106.99_{\pm0.48}$ | $104.08_{\pm3.64}$ | $104.94_{\pm3.17}$ | $99.36_{\pm5.90}$ |
| | $PIS_\theta + TLM_\varphi$ | (learnt var.) | $106.95_{\pm0.42}$ | $106.95_{\pm0.44}$ | $106.95_{\pm0.42}$ | $106.99_{\pm0.37}$ | $106.98_{\pm0.40}$ | $107.00_{\pm0.40}$ | $99.51_{\pm11.05}$ | $105.41_{\pm2.71}$ |
| | $PIS_\theta + VarGrad_\varphi$ | (learnt var.) | $106.95_{\pm0.42}$ | $106.95_{\pm0.44}$ | $106.95_{\pm0.42}$ | $106.99_{\pm0.37}$ | $106.98_{\pm0.40}$ | $102.54_{\pm5.94}$ | $106.51_{\pm1.12}$ | $104.24_{\pm4.27}$ |

### 2-Wasserstein ($\downarrow$)

| Energy $\downarrow$ | Method $\downarrow$ Steps $\rightarrow$ | | $T=2$ | $T=3$ | $T=4$ | $T=5$ | $T=10$ | $T=15$ | $T=20$ | $T=25$ |
|---|---|---|---|---|---|---|---|---|---|---|
| 40GMM | $TB_\theta$ | (fixed var.) | $38.72_{\pm5.09}$ | $33.46_{\pm2.17}$ | $33.93_{\pm2.24}$ | $31.12_{\pm2.34}$ | $29.18_{\pm1.39}$ | $29.93_{\pm0.16}$ | $27.91_{\pm2.44}$ | $29.97_{\pm0.63}$ |
| | $TB_\theta + TLM_\varphi$ | (fixed var.) | $34.00_{\pm2.88}$ | $29.13_{\pm0.14}$ | $29.27_{\pm0.15}$ | $29.19_{\pm0.08}$ | $28.28_{\pm0.13}$ | $27.45_{\pm0.36}$ | $25.74_{\pm1.76}$ | $24.94_{\pm2.66}$ |
| | $TB_{\theta,\varphi}$ | (fixed var.) | $31.65_{\pm2.70}$ | $29.21_{\pm0.09}$ | $28.97_{\pm0.27}$ | $28.42_{\pm0.57}$ | $27.00_{\pm0.53}$ | $26.84_{\pm0.65}$ | $26.21_{\pm1.33}$ | $25.54_{\pm1.74}$ |
| | $TB_\theta$ | (learnt var.) | $18.36_{\pm0.71}$ | $20.59_{\pm2.00}$ | $16.34_{\pm1.75}$ | $21.99_{\pm6.57}$ | $15.63_{\pm2.28}$ | $15.14_{\pm1.33}$ | $14.02_{\pm0.28}$ | $19.38_{\pm7.77}$ |
| | $TB_\theta + TLM_\varphi$ | (learnt var.) | $10.02_{\pm0.23}$ | $12.10_{\pm2.73}$ | $15.53_{\pm0.71}$ | $19.84_{\pm1.53}$ | $18.85_{\pm0.74}$ | $18.08_{\pm1.74}$ | $19.64_{\pm1.49}$ | $20.98_{\pm1.49}$ |
| | $TB_{\theta,\varphi}$ | (learnt var.) | $10.79_{\pm0.39}$ | $11.47_{\pm1.38}$ | $11.29_{\pm1.60}$ | $19.37_{\pm1.51}$ | $16.45_{\pm0.79}$ | $16.18_{\pm0.61}$ | $16.82_{\pm1.22}$ | $17.83_{\pm1.77}$ |
| | $PIS_\theta$ | (fixed var.) | $30.44_{\pm0.07}$ | $30.45_{\pm0.05}$ | $30.44_{\pm0.08}$ | $30.43_{\pm0.07}$ | $30.46_{\pm0.04}$ | $30.47_{\pm0.09}$ | $30.46_{\pm0.05}$ | $30.46_{\pm0.06}$ |
| | $PIS_\theta$ | (learnt var.) | $30.43_{\pm0.07}$ | $30.37_{\pm0.14}$ | $30.15_{\pm0.18}$ | $30.09_{\pm0.28}$ | $30.09_{\pm0.21}$ | $30.15_{\pm0.35}$ | $30.14_{\pm0.45}$ | $29.73_{\pm0.60}$ |
| | $PIS_\theta + TLM_\varphi$ | (learnt var.) | $30.22_{\pm0.35}$ | $30.14_{\pm0.21}$ | $29.93_{\pm0.57}$ | $29.92_{\pm0.07}$ | $29.63_{\pm0.27}$ | $29.91_{\pm0.26}$ | $29.85_{\pm0.17}$ | $29.86_{\pm0.34}$ |
| | $PIS_\theta + VarGrad_\varphi$ | (learnt var.) | $28.33_{\pm3.00}$ | $30.42_{\pm0.08}$ | $30.08_{\pm0.25}$ | $30.06_{\pm0.50}$ | $30.46_{\pm0.06}$ | $30.15_{\pm0.49}$ | $30.22_{\pm0.24}$ | $30.16_{\pm0.39}$ |
| | Ground Truth | | | | | $3.95_{\pm0.51}$ | | | | |

Table 9: Comparison of 4 algorithms by ELBO, EUBO, and 2-Wasserstein between generated and ground truth samples with varying number of discretisation steps on Easy Funnel and Hard Funnel. Mean and std over 3 runs are specified.

### ELBO ($\uparrow$)

| Energy $\downarrow$ | Method $\downarrow$ Steps $\rightarrow$ | | $T = 5$ | $T = 10$ | $T = 15$ | $T = 20$ | $T = 25$ | $T = 30$ |
|---|---|---|---|---|---|---|---|---|
| Easy Funnel | $TB_\theta$ | (fixed var.) | $-0.61_{\pm 0.00}$ | $-0.34_{\pm 0.02}$ | $-0.23_{\pm 0.02}$ | $-0.19_{\pm 0.01}$ | $-0.15_{\pm 0.00}$ | $-0.14_{\pm 0.00}$ |
| | $TB_\theta + TLM_\varphi$ | (fixed var.) | $-0.58_{\pm 0.02}$ | $-0.31_{\pm 0.01}$ | diverged | diverged | diverged | diverged |
| | $TB_{\theta,\varphi}$ | (fixed var.) | $-0.62_{\pm 0.03}$ | $-0.33_{\pm 0.01}$ | $-0.22_{\pm 0.01}$ | $-0.17_{\pm 0.00}$ | $-0.16_{\pm 0.01}$ | $-0.13_{\pm 0.01}$ |
| | $TB_\theta$ | (learnt var.) | $-0.34_{\pm 0.01}$ | $-0.17_{\pm 0.01}$ | $-0.11_{\pm 0.00}$ | $-0.08_{\pm 0.01}$ | $-0.07_{\pm 0.01}$ | $-0.07_{\pm 0.01}$ |
| | $TB_\theta + TLM_\varphi$ | (learnt var.) | $-0.08_{\pm 0.04}$ | $-0.11_{\pm 0.01}$ | diverged | diverged | diverged | diverged |
| | $TB_{\theta,\varphi}$ | (learnt var.) | $-0.15_{\pm 0.01}$ | $-0.11_{\pm 0.01}$ | $-0.08_{\pm 0.00}$ | $-0.06_{\pm 0.01}$ | $-0.05_{\pm 0.00}$ | $-0.05_{\pm 0.00}$ |
| Hard Funnel | $TB_\theta$ | (fixed var.) | $-1.68_{\pm 0.02}$ | $-1.17_{\pm 0.03}$ | $-0.98_{\pm 0.01}$ | $-0.89_{\pm 0.02}$ | $-0.79_{\pm 0.00}$ | $-0.76_{\pm 0.01}$ |
| | $TB_\theta + TLM_\varphi$ | (fixed var.) | $-1.43_{\pm 0.01}$ | $-1.09_{\pm 0.03}$ | diverged | diverged | diverged | diverged |
| | $TB_{\theta,\varphi}$ | (fixed var.) | $-1.83_{\pm 0.08}$ | $-1.58_{\pm 0.45}$ | $-0.95_{\pm 0.01}$ | $-0.85_{\pm 0.02}$ | $-0.79_{\pm 0.03}$ | $-0.76_{\pm 0.01}$ |
| | $TB_\theta$ | (learnt var.) | $-1.72_{\pm 0.04}$ | $-1.25_{\pm 0.03}$ | $-0.91_{\pm 0.05}$ | $-0.61_{\pm 0.01}$ | $-0.48_{\pm 0.02}$ | $-0.44_{\pm 0.01}$ |
| | $TB_\theta + TLM_\varphi$ | (learnt var.) | $-0.63_{\pm 0.26}$ | $-0.45_{\pm 0.04}$ | diverged | diverged | diverged | diverged |
| | $TB_{\theta,\varphi}$ | (learnt var.) | $-1.01_{\pm 0.04}$ | $-0.69_{\pm 0.04}$ | $-0.56_{\pm 0.03}$ | $-0.55_{\pm 0.08}$ | $-0.41_{\pm 0.02}$ | $-0.41_{\pm 0.03}$ |

### EUBO ($\downarrow$)

| Energy $\downarrow$ | Method $\downarrow$ Steps $\rightarrow$ | | $T = 5$ | $T = 10$ | $T = 15$ | $T = 20$ | $T = 25$ | $T = 30$ |
|---|---|---|---|---|---|---|---|---|
| Easy Funnel | $TB_\theta$ | (fixed var.) | $0.73_{\pm 0.01}$ | $0.40_{\pm 0.02}$ | $0.28_{\pm 0.01}$ | $0.21_{\pm 0.01}$ | $0.17_{\pm 0.00}$ | $0.15_{\pm 0.00}$ |
| | $TB_\theta + TLM_\varphi$ | (fixed var.) | $0.64_{\pm 0.02}$ | $0.35_{\pm 0.01}$ | $2.80_{\pm 0.11}$ | diverged | diverged | diverged |
| | $TB_{\theta,\varphi}$ | (fixed var.) | $0.84_{\pm 0.03}$ | $0.39_{\pm 0.04}$ | $0.26_{\pm 0.01}$ | $0.20_{\pm 0.01}$ | $0.18_{\pm 0.01}$ | $0.15_{\pm 0.02}$ |
| | $TB_\theta$ | (learnt var.) | $0.38_{\pm 0.02}$ | $0.18_{\pm 0.01}$ | $0.12_{\pm 0.00}$ | $0.09_{\pm 0.00}$ | $0.08_{\pm 0.00}$ | $0.06_{\pm 0.00}$ |
| | $TB_\theta + TLM_\varphi$ | (learnt var.) | $0.08_{\pm 0.03}$ | $0.12_{\pm 0.01}$ | diverged | diverged | diverged | diverged |
| | $TB_{\theta,\varphi}$ | (learnt var.) | $0.17_{\pm 0.03}$ | $0.12_{\pm 0.01}$ | $0.08_{\pm 0.00}$ | $0.07_{\pm 0.01}$ | $0.05_{\pm 0.00}$ | $0.05_{\pm 0.00}$ |
| Hard Funnel | $TB_\theta$ | (fixed var.) | $95.36_{\pm 8.41}$ | $78.77_{\pm 8.35}$ | $75.40_{\pm 3.35}$ | $72.83_{\pm 5.15}$ | $70.99_{\pm 4.26}$ | $68.93_{\pm 5.01}$ |
| | $TB_\theta + TLM_\varphi$ | (fixed var.) | $109.52_{\pm 52.22}$ | $55.76_{\pm 17.37}$ | $285.59_{\pm 53.98}$ | diverged | diverged | diverged |
| | $TB_{\theta,\varphi}$ | (fixed var.) | $374.26_{\pm 228.11}$ | $694.95_{\pm 901.63}$ | $79.03_{\pm 50.28}$ | $69.47_{\pm 30.67}$ | $56.34_{\pm 26.13}$ | $95.48_{\pm 24.76}$ |
| | $TB_\theta$ | (learnt var.) | $155.51_{\pm 19.60}$ | $102.97_{\pm 41.86}$ | $105.67_{\pm 6.26}$ | $121.76_{\pm 41.45}$ | $349.17_{\pm 406.05}$ | $79.37_{\pm 33.30}$ |
| | $TB_\theta + TLM_\varphi$ | (learnt var.) | $1509.44_{\pm 2037.59}$ | $22.77_{\pm 14.77}$ | diverged | diverged | diverged | diverged |
| | $TB_{\theta,\varphi}$ | (learnt var.) | $2800.63_{\pm 1351.92}$ | $21.48_{\pm 13.09}$ | $62.95_{\pm 75.95}$ | $30.35_{\pm 20.62}$ | $8.60_{\pm 0.97}$ | $17.29_{\pm 8.63}$ |

### 2-Wasserstein ($\downarrow$)

| Energy $\downarrow$ | Method $\downarrow$ Steps $\rightarrow$ | | $T = 5$ | $T = 10$ | $T = 15$ | $T = 20$ | $T = 25$ | $T = 30$ |
|---|---|---|---|---|---|---|---|---|
| Easy Funnel | $TB_\theta$ | (fixed var.) | $2.45_{\pm 0.04}$ | $2.43_{\pm 0.03}$ | $2.45_{\pm 0.04}$ | $2.49_{\pm 0.01}$ | $2.47_{\pm 0.02}$ | $2.48_{\pm 0.03}$ |
| | $TB_\theta + TLM_\varphi$ | (fixed var.) | $2.44_{\pm 0.02}$ | $2.45_{\pm 0.03}$ | $2.60_{\pm 0.03}$ | diverged | diverged | diverged |
| | $TB_{\theta,\varphi}$ | (fixed var.) | $2.45_{\pm 0.04}$ | $2.45_{\pm 0.01}$ | $2.45_{\pm 0.01}$ | $2.47_{\pm 0.02}$ | $2.49_{\pm 0.03}$ | $2.49_{\pm 0.03}$ |
| | $TB_\theta$ | (learnt var.) | $2.49_{\pm 0.02}$ | $2.50_{\pm 0.03}$ | $2.49_{\pm 0.03}$ | $2.49_{\pm 0.02}$ | $2.50_{\pm 0.02}$ | $2.53_{\pm 0.01}$ |
| | $TB_\theta + TLM_\varphi$ | (learnt var.) | $2.51_{\pm 0.05}$ | $2.53_{\pm 0.02}$ | diverged | diverged | diverged | diverged |
| | $TB_{\theta,\varphi}$ | (learnt var.) | $2.52_{\pm 0.01}$ | $2.49_{\pm 0.02}$ | $2.51_{\pm 0.03}$ | $2.53_{\pm 0.02}$ | $2.52_{\pm 0.02}$ | $2.50_{\pm 0.02}$ |
| | Ground Truth | | $2.58_{\pm 0.05}$ | | | | | |
| Hard Funnel | $TB_\theta$ | (fixed var.) | $22.82_{\pm 0.73}$ | $22.50_{\pm 0.75}$ | $22.36_{\pm 0.78}$ | $22.17_{\pm 0.81}$ | $22.08_{\pm 0.79}$ | $21.99_{\pm 0.79}$ |
| | $TB_\theta + TLM_\varphi$ | (fixed var.) | $22.43_{\pm 2.39}$ | $22.05_{\pm 2.41}$ | $23.07_{\pm 2.36}$ | diverged | diverged | diverged |
| | $TB_{\theta,\varphi}$ | (fixed var.) | $22.53_{\pm 2.39}$ | $22.47_{\pm 2.50}$ | $21.90_{\pm 2.40}$ | $21.53_{\pm 2.53}$ | $21.26_{\pm 2.48}$ | $21.18_{\pm 2.52}$ |
| | $TB_\theta$ | (learnt var.) | $22.45_{\pm 0.79}$ | $21.94_{\pm 0.85}$ | $21.63_{\pm 0.80}$ | $21.53_{\pm 0.76}$ | $21.44_{\pm 0.81}$ | $21.36_{\pm 0.81}$ |
| | $TB_\theta + TLM_\varphi$ | (learnt var.) | $21.32_{\pm 0.90}$ | $21.04_{\pm 0.84}$ | diverged | diverged | diverged | diverged |
| | $TB_{\theta,\varphi}$ | (learnt var.) | $21.48_{\pm 0.81}$ | $20.96_{\pm 0.88}$ | $21.04_{\pm 0.91}$ | $20.91_{\pm 0.86}$ | $21.03_{\pm 0.80}$ | $21.01_{\pm 0.90}$ |
| | Ground Truth | | $24.24_{\pm 4.20}$ | | | | | |

Table 10: Comparison of 4 algorithms by ELBO gap, EUBO gap, and 2-Wasserstein between generated and ground truth samples with varying number of discretisation steps on Manywell and Distorted Manywell. Mean and std over 3 runs are specified.

| | ELBO gap ($\uparrow$) | | | |
|---|---|---|---|---|
| Energy $\downarrow$ | Method $\downarrow$ Steps $\rightarrow$ | $T = 5$ | $T = 10$ | $T = 20$ |
| ManyWell | $\text{TB}_\theta$ (fixed var.) | $-36.25_{\pm13.68}$ | $-12.36_{\pm0.10}$ | $-5.67_{\pm0.05}$ |
| | $\text{TB}_\theta$ (learnt var.) | $-2.40_{\pm0.03}$ | $-0.78_{\pm0.02}$ | $-0.41_{\pm0.09}$ |
| | $\text{TB}_\theta + \text{TLM}_\varphi$ | $-2.16_{\pm0.12}$ | $-0.79_{\pm0.01}$ | diverged |
| | $\text{TB}_{\theta,\varphi}$ | $-2.22_{\pm0.05}$ | $-0.83_{\pm0.02}$ | $-0.32_{\pm0.01}$ |
| Distorted ManyWell | $\text{TB}_\theta$ (fixed var.) | $-77.74_{\pm6.27}$ | $-25.75_{\pm9.45}$ | $-11.96_{\pm0.16}$ |
| | $\text{TB}_\theta$ (learnt var.) | $-8.36_{\pm0.73}$ | $-5.70_{\pm1.04}$ | $-3.83_{\pm0.13}$ |
| | $\text{TB}_\theta + \text{TLM}_\varphi$ | $-6.47_{\pm1.12}$ | $-3.99_{\pm2.31}$ | diverged |
| | $\text{TB}_{\theta,\varphi}$ | $-7.33_{\pm1.34}$ | $-4.07_{\pm2.38}$ | $-2.90_{\pm1.60}$ |

| | EUBO gap ($\downarrow$) | | | |
|---|---|---|---|---|
| Energy $\downarrow$ | Method $\downarrow$ Steps $\rightarrow$ | $T = 5$ | $T = 10$ | $T = 20$ |
| ManyWell | $\text{TB}_\theta$ (fixed var.) | $12.80_{\pm3.02}$ | $6.72_{\pm0.02}$ | $4.06_{\pm0.04}$ |
| | $\text{TB}_\theta$ (learnt var.) | $1.68_{\pm0.02}$ | $0.63_{\pm0.01}$ | $0.36_{\pm0.09}$ |
| | $\text{TB}_\theta + \text{TLM}_\varphi$ | $1.69_{\pm0.01}$ | $0.72_{\pm0.02}$ | diverged |
| | $\text{TB}_{\theta,\varphi}$ | $1.61_{\pm0.01}$ | $0.64_{\pm0.01}$ | $0.31_{\pm0.01}$ |
| Distorted ManyWell | $\text{TB}_\theta$ (fixed var.) | $19.68_{\pm0.62}$ | $10.46_{\pm2.30}$ | $6.19_{\pm0.10}$ |
| | $\text{TB}_\theta$ (learnt var.) | $4.53_{\pm0.30}$ | $3.16_{\pm0.37}$ | $2.15_{\pm0.08}$ |
| | $\text{TB}_\theta + \text{TLM}_\varphi$ | $4.09_{\pm0.49}$ | $2.39_{\pm1.09}$ | diverged |
| | $\text{TB}_{\theta,\varphi}$ | $4.29_{\pm0.42}$ | $2.38_{\pm1.19}$ | $1.77_{\pm0.80}$ |

| | 2-Wasserstein ($\downarrow$) | | | |
|---|---|---|---|---|
| Energy $\downarrow$ | Method $\downarrow$ Steps $\rightarrow$ | $T = 5$ | $T = 10$ | $T = 20$ |
| ManyWell | $\text{TB}_\theta$ (fixed var.) | $5.57_{\pm0.11}$ | $5.40_{\pm0.02}$ | $5.38_{\pm0.01}$ |
| | $\text{TB}_\theta$ (learnt var.) | $5.35_{\pm0.01}$ | $5.38_{\pm0.01}$ | $5.41_{\pm0.03}$ |
| | $\text{TB}_\theta + \text{TLM}_\varphi$ | $5.33_{\pm0.03}$ | $5.36_{\pm0.02}$ | diverged |
| | $\text{TB}_{\theta,\varphi}$ | $5.36_{\pm0.02}$ | $5.39_{\pm0.01}$ | $5.40_{\pm0.01}$ |
| | Ground Truth | | $5.42_{\pm0.02}$ | |
| Distorted ManyWell | $\text{TB}_\theta$ (fixed var.) | $5.87_{\pm0.03}$ | $5.66_{\pm0.02}$ | $5.53_{\pm0.06}$ |
| | $\text{TB}_\theta$ (learnt var.) | $5.53_{\pm0.00}$ | $5.44_{\pm0.03}$ | $5.40_{\pm0.01}$ |
| | $\text{TB}_\theta + \text{TLM}_\varphi$ | $5.50_{\pm0.02}$ | $5.42_{\pm0.09}$ | diverged |
| | $\text{TB}_{\theta,\varphi}$ | $5.54_{\pm0.01}$ | $5.41_{\pm0.08}$ | $5.37_{\pm0.06}$ |
| | Ground Truth | | $5.30_{\pm0.01}$ | |

Table 11: Comparison of 4 algorithms by ELBO with varying number of discretisation steps on VAE. Mean and std over 3 runs are specified.

| | ELBO ($\uparrow$) | | | | |
|---|---|---|---|---|---|
| Energy $\downarrow$ | Method $\downarrow$ Steps $\rightarrow$ | $T = 4$ | $T = 8$ | $T = 16$ | $T = 32$ | $T = 64$ |
| VAE | $\text{VarGrad}_\theta$ (fixed var.) | $-168.51_{\pm3.97}$ | $-144.97_{\pm0.88}$ | $-131.55_{\pm0.86}$ | $-122.78_{\pm0.72}$ | $-117.30_{\pm3.37}$ |
| | $\text{VarGrad}_\theta$ (learnt var.) | $-117.00_{\pm1.00}$ | $-113.13_{\pm0.69}$ | $-111.34_{\pm1.97}$ | $-110.43_{\pm2.18}$ | $-116.12_{\pm4.17}$ |
| | $\text{VarGrad}_\theta + \text{TLM}_\varphi$ | $-104.00_{\pm0.14}$ | $-105.27_{\pm0.94}$ | $-105.23_{\pm0.70}$ | $-105.39_{\pm0.85}$ | $-104.56_{\pm0.46}$ |
| | $\text{VarGrad}_{\theta,\varphi}$ | $-107.86_{\pm3.93}$ | $-106.46_{\pm1.66}$ | $-105.07_{\pm0.83}$ | $-103.89_{\pm1.11}$ | $-104.10_{\pm0.35}$ |

