# OpenReview forum: "Adaptive Destruction Processes for Diffusion Samplers"
_ICLR.cc/2026/Conference — Submitted to ICLR 2026_

### Official Review · Reviewer_83Ld · 2025-10-27

**Soundness:** 3
**Presentation:** 3
**Contribution:** 3
**Rating:** 8
**Confidence:** 3

**Summary:**

The paper jointly learns generation and destruction in a discrete-time sampler, enabling state-dependent, learnable variance in both directions. This sidesteps continuous-time path-KL issues when variances differ and is supported by bounded parameterizations plus stabilization tactics (shared backbone, separate optimizers, target nets, replay). Empirically, it helps most in few-step regimes and narrow-mode targets; second-moment objectives (TB/VarGrad) are competitive with or better than PIS while being more memory-friendly. A style-transfer/latent-space demo shows the approach scales. Main trade-offs: weaker ties to continuous-time bridges, hyperparameter sensitivity (especially destruction LR), occasional instability (e.g., TLM at large steps), limited breadth beyond faces, and extra compute for some conditional setups.

**Strengths:**

1. Clear motivation; concrete contribution: learnable, state-dependent variance for both directions in discrete time.

2. Thoughtful engineering for stability with solid ablations.

3. Good theoretical positioning vs. IPF/SB/CMCD/GFlowNet.

**Weaknesses:**

1. Continuous-time connection & guarantees. Because the method allows different variances for generation vs. destruction, the link to continuous-time bridge formalisms seems less direct. Could you clarify what guarantees still hold in discrete time (e.g., well-posedness, stability, convergence/consistency), and how you ensure or demonstrate them in practice? Any safeguards to prevent pathological transitions when the two variances diverge?

2. Hyperparameter sensitivity & tuning. The approach appears sensitive to learning-rate ratios—especially for the destruction model—which may affect stability and reproducibility across datasets/seeds.  How to guarantee the stability of training across settings?

**Questions:**

See the weaknesses.

---

> ### Author Response · Authors · 2025-11-21
>
> We sincerely thank the reviewer for their positive assessment of our work. We are grateful that you highlight the clarity of the motivation, the concrete contribution of jointly learning state-dependent variances in both directions, the careful engineering for stability with supporting ablations, and the positioning of our approach with respect to other methods.
>
> We now proceed to the reviewer's concerns.
>
> > The approach appears sensitive to learning-rate ratios—especially for the destruction model [...] How to guarantee the stability of training across settings?
>
> This is indeed an important question. While the choice of the destruction model’s learning rate is crucial for stability, it does not inherently prevent reproducibility. Our code produces reproducible results when the random seed is fixed.
>
> To support stable training across settings, we systematically ablate the proposed design choices in §4.3 and fix a single hyperparameter configuration for all experiments in §4.2, varying only the destruction learning rate.
>
> > Continuous-time connection & guarantees. Because the method allows different variances for generation vs. destruction, the link to continuous-time bridge formalisms seems less direct. Could you clarify what guarantees still hold in discrete time (e.g., well-posedness, stability, convergence/consistency), and how you ensure or demonstrate them in practice? Any safeguards to prevent pathological transitions when the two variances diverge?
>
> As far as we know, learning guarantees have never been studied for diffusion samplers neither in continuous, nor in discrete time. Furthermore, stability guarantees are generally tied to the approximating function / distribution class (e.g., Gaussian transition kernels with mean and variance given by neural networks of a certain architecture). However, we have some thoughts about the question.
>
> **In continuous time:** It is true that in continuous time, SDEs with sufficiently regular coefficients have unique reverse SDEs and thus unique loss minimisers within the class of all sufficiently regular SDEs (was this the well-posedness you referred to?).
>
> The continuous-time limiting behaviour of discrete-time objectives was most generally studied in [Berner et al., 2025], but this analysis is only applicable when the variances of the generation and destruction processes coincide, since otherwise the continuous-time limits of the training objectives does not exist (the limiting processes are defined by SDEs with different diffusion coefficients and are not absolutely continuous with respect to one another). In fact, the learnt variances of the generation and destruction processes, which we study here, are tied to the choice of fixed time discretisation and motivated by the need to adapt to this choice, where simple Euler-Maruyama integration of SDEs with fixed variances is suboptimal.
>
> **In discrete time:** Once we have fixed a time discretisation, the objectives become divergences between discrete-time Markov chains, which in our case are distributions over finite-dimensional spaces. It is still true that any destruction kernel $\overleftarrow p$ has a unique reverse kernel. However, this reverse will not, in general, be Gaussian if the destruction process is Gaussian, meaning that the loss cannot be minimised to 0.
>
> This failure to perfectly reverse a Gaussian destruction process by a generation process with Gaussian transitions is a general limitation of training and sampling from an underlyingly continuous-time model in discrete time. In typical diffusion models trained on a dataset, the error is incurred at sampling time, through numerical integration. In training of diffusion sampler, the error additionally appears during training because we must simulate trajectories from a behaviour policy; however, as we show in this paper, allowing the kernels to learn to adapt to the chosen discretisation (for example, by learning the variances) can reduce this error.
>
> To conclude, we note that one of the objectives considered in this work is the (trajectory-level) reverse KL,
> $D_{\mathrm{KL}}\big(p_0  p_\theta^{\rightarrow} \big\| p_{\mathrm{target}} p_\varphi^{\leftarrow}\big),$ which upper-bounds the KL between the target and modelled distributions:
> $$D_{\mathrm{KL}}\big(p_\theta^{\rightarrow}(X_1) \big\| p_{\mathrm{target}}(X_1)\big)
> \le
> D_{\mathrm{KL}}\big(p_0(X_0) p_\theta^{\rightarrow}(X_{0:1}\mid X_0) \big \|
> p_{\mathrm{target}}(X_1)\, p_\varphi^{\leftarrow}(X_{1:0}\mid X_1)\big).$$
> This holds for any choice of $\overleftarrow{p_\varphi}$, whether fixed or learnt. The PIS and on-policy TB objectives both give **unbiased** estimates of the reverse KL gradient with respect to $\theta$, so we are in the setting of standard stochastic gradient descent, where stability guarantees depend on smoothness of the loss near the optimum.
>
> **Thank you again for your positive review, we hope we have clarified the questions you raised.**

---

### Official Review · Reviewer_YXt4 · 2025-10-29

**Soundness:** 2
**Presentation:** 2
**Contribution:** 2
**Rating:** 4
**Confidence:** 3

**Summary:**

The paper proposed a trainable destruction process technique during discrete diffusion model's sampling process. In detail, the method proposed a novel view for diffusion sampling. The experimental results showed that the deconstruction process yielded in fastering convergence, and higher ELBO. And the ablation studies show the effectiveness of the decomposition process.

**Strengths:**

1. Framework novelty. The method firstly extend the traditional diffusion process into learnable variances in an unified theoretical framework.

2. Integration of stability mechanism. The paper involved reinforcement-learning stabilization tools inspired by reinforcement learning's view. And Table 2 systematically evaluate the performance of each tool.

3. Scalability to high-dimensional tasks. Section 4.4 demonstrated the capability of the method to higher dimension image generation tasks, which leads to boarder applications.

**Weaknesses:**

1. Insufficient theoretical analysis. Although there is unified framework and well-defined processes, no analysis of the convergence or gradient bias of KL divergence is provided.

2. Lack of continuous-time analysis. There is no proof for the equivalence between the generation and th destruction processes as T goes to infinity.

3. Limited evaluation to TLM. The paper proposed TB and TLM, but the main experiments were conducted by TB.

**Questions:**

1. In equation 14, does the choice of proposal distribution tied to the off-policy exploration in Section 3.3?

2. Is there analysis showing the connection between the learnable variances between two processes?

3. How will the method behaive if the energy landscape violates smoothness assumptions (non-Lipschitz E(x))? Does the KL objective in Eq. (13) remain well-defined under this situation?

4. Were target networks and PER ablated individually or only in combination (Table 2)? How sensitive is performance to the PER replay ratio?

5. The paper alternates between KL-based and second-moment objectives. Is there a principled reason not to use hybrid losses (weighted KL + TB)?

---

> ### Author Response · Authors · 2025-11-21
>
> We thank the reviewer for their careful reading of our paper and for highlighting several strengths of our work. We appreciate that they recognize the novelty of our unified framework with learnable variances, the integration of stability mechanisms inspired by reinforcement learning (and the systematic evaluation in Table 2), as well as the demonstrated scalability of our approach to high-dimensional image generation tasks in §4.4. In the revision, we further strengthen this point by adding additional results on StyleGAN3, including more visual examples, to better illustrate scalability and sampling quality.
>
> We proceed to answer the reviewer's concerns.
>
> > Limited evaluation to TLM. The paper proposed TB and TLM, but the main experiments were conducted by TB.
>
> In nearly all past works in the literature on GFlowNets, from which our algorithms are inspired, where both generation and destruction policies are trained, TB training is used for both. **The version of our proposed algorithm that uses TB for both processes should be considered the "base" version, and it is stable across all experiments.**
>
> An exception is [Gritsaev et al., 2025], which studied discrete-space problems, proposing TLM as an alternative method for training the destruction process. We included TLM for completeness, with the goal of seeing whether the findings of [Gritsaev et al.] generalise to the diffusion sampling case. We find that for short trajectories TLM works well, but for long ones it is unstable.
>
> We also emphasize that experiments with TLM were conducted on all tasks in §4.1. (Perhaps it was not made clear enough that TLM refers only to the use of Reverse KL to train the destruction process and cannot be used to train the generation process, and we will make sure to state this in the text.) Results for both TB and TLM are shown in Figures 1 and 2. **In addition, we now provide new quantitative results on StyleGAN3, with full numbers including TLM reported in the table below.** We stress that the prior is an unconditional model. After this update, results for TLM are provided across all experiments.
>
> |Algorithm|ELBO|Reward|CLIP diversity|
> |---|---|---|---|
> |Prior (unconditional)|-190|-1.90|0.362|
> |**(Baseline)** TB$_\\theta$ [Venkatraman et al.]|20.8|0.988|0.246|
> |**(Ours)** TB$_\\theta$ (learnt var.)|43.7|1.10|0.235|
> |**(Ours)** TB$_{\\theta,\\varphi}$|53.7|1.23|0.231|
> |**(Ours)** TB$_\\theta$|59.5|1.27|0.230|
>
> > Insufficient theoretical analysis. Although there is unified framework and well-defined processes, no analysis of the convergence or gradient bias of KL divergence is provided.
>
> We are unsure how to intepret this question: what gradient bias is being referred to?
>
> Considering the objectives summarised in Table 1 that optimise KL with respect to the generation process, the gradient of the objective in the last column (PIS$_\theta$) is trivially an **unbiased** estimator of the KL gradient, since it expresses the KL in equation (13) as a function of exogenous noise (reparametrisation trick) and places the gradient within the expectation. The gradient of the second-moment divergence is also an **unbiased** estimator of the KL gradient when optimised on policy, a result known in the literature that we mention in the discussion between equations (14) and (15).
>
> If we have misunderstood the question, we kindly ask for clarification.
>
> (continued in comment below)

---

> > ### Author Response · Authors · 2025-11-21
> >
> > (continued from comment above)
> >
> >
> > > Lack of continuous-time analysis. There is no proof for the equivalence between the generation and th destruction processes as T goes to infinity.
> >
> > In the case where the variances of the generation and destruction processes coincide and depend only the time variable, the continuous-time limiting behaviour was studied in [Berner et al., "From discrete-time policies...", 2025]. This paper proved that in the continuous-time limit, discrete-time training objectives, including Reverse KL (PIS) and TB, approach divergences between path space measures, implying that **integration in continuous time would sample from the target distribution when the loss is minimised in the limit**. We note that while that paper's experimental section only used fixed (pinned Brownian motion) destruction processes, the theoretical analysis did not assume any particular destruction process and is thus applicable to our case.
> >
> > However, **the need to learn the variances of the generation and destruction processes emerges from the choice of fixed time discretisation, and learnt variances adapt to this choice.** When it is not enforced that the variances coincide, the limit of the training objectives does not exist -- indeed, the limiting distributions are processes defined by SDEs that are not absolutely continuous with respect to one another. Thus it does not make sense to take the number of discretisation steps to infinity in this case.
> >
> > > In equation 14, does the choice of proposal distribution tied to the off-policy exploration in Section 3.3?
> >
> > Yes, $\overset{\sim}{p}(X)$ is some full-support proposal distribution not necessarily equal to $\overset{\rightarrow}{p}(X)$, for example, one sampled by an off-policy exploration strategy.
> >
> > > Were target networks and PER ablated individually or only in combination (Table 2)? How sensitive is performance to the PER replay ratio?
> >
> > We aimed to demonstrate that each design choice is important and justified. Accordingly, our ablation study includes a base version that uses all stabilization techniques, plus variants where we remove exactly one component at a time. Target networks and replay ratio were therefore ablated individually. When varying the replay ratio with target networks disabled, we obtained slightly worse results compared to the configuration with target networks and the same replay ratio setting.
> >
> > > The paper alternates between KL-based and second-moment objectives. Is there a principled reason not to use hybrid losses (weighted KL + TB)?
> >
> > No, there is no reason the losses could not be combined. KL-based and second-moment objectives are standard choices in prior work, proposed in [Zhang et al., "Path integral sampler...", 2022] and [Malkin et al, "Trajectory balance...", 2022], [Richter et al., "VarGrad...", 2020], respectively. Our goal in this paper is to demonstrate how to train the destruction process using these well-established objectives. Hybrid losses (e.g., weighted combinations of KL and TB) could indeed work well in practice, but tuning such combinations is not the point of the present work.
> >
> > > Is there analysis showing the connection between the learnable variances between two processes?
> >
> > We are not aware of any such analysis in discrete time (i.e., in the few-step case). Even for a one-step sampler, the question reduces to one of finding the optimal Gaussian approximation to a given distribution in the reverse KL sense, which depends on the target distribution in a way that is not analytically tractable in general.
> >
> > In the continuous-time limit, the two variances should coincide, since the reverse process to an Euler-Maruyama-discretised SDE approaches the discretisation of its reverse SDE.
> >
> > > How will the method behaive if the energy landscape violates smoothness assumptions (non-Lipschitz E(x))? Does the KL objective in Eq. (13) remain well-defined under this situation?
> >
> > Firstly, we do not see why non-Lipschitzness of $\cal E$ should be a problem. In fact, a standard normal distribution is the Boltzmann distribution of the non-Lipschitz function $\frac{\|x\|^2}{2}$.
> >
> > Secondly, the KL is defined as long as the generation and destruction processes are absolutely continuous with respect to one another. In our case, the generation process is defined by a distribution over $X_0$ together with a conditionally Gaussian generation kernel, and the destruction process is defined by a full-support distribution over $X_1$ (the Boltzmann distribution with density $\exp(-\cal E(x))$) together with a conditionally Gaussian destruction kernel. Absolute continuity holds under very basic conditions that are always true in practice (e.g., if $\exp(-{\cal E}(x))$ is integrable and the conditional means and variances of the kernels are measurable functions). Note that there is no requirement of local smoothness (e.g., locally Lipschitz).
> >
> > **Thank you again for your review, and do not hesitate to let us know if you have further questions.**

---

> > > ### Comment · Reviewer_YXt4 · 2025-11-26
> > >
> > > Dear authors,
> > >
> > > Thanks for your rebuttal! Part of the responses sounds good to me. Here is my following question:
> > >
> > > I believe the rebuttal content didnot resolve the fundamental issue that the proposed training setup is strongly off-policy, with $\(\tilde p \neq p_\theta\)$, while using PER and target networks inducing non-stationary targets.
> > >
> > > In this regime, the authors' claim "unbiased KL-related gradients doesnot holds". However, no theoretical bounds or empirical diagnostics are provided. Thus, it remains unclear if the reported performance gains reflect true KL optimization or come from RL stabilisation.
> > >
> > > Best,
> > >
> > > Reviewer

---

> > > > ### Author Response · Authors · 2025-11-26
> > > > **on off-policy methods and KL gradients**
> > > >
> > > > Thank you for your follow-up question, which we now respond to, first by addressing the theoretical/algorithmic aspects, then by recalling the empirical evidence.
> > > >
> > > > Regarding theory and methodology:
> > > > 1. **An important clarification:** In our work, we study both on-policy and off-policy training schemes: on-policy PIS [Zhang et al., "Path Integral Sampler...", 2022], and off-policy TB [Malkin et al., "Trajectory Balance...", 2022] and VarGrad [Richter et al., "VarGrad...", 2020]. **We use a prioritised replay buffer only for off-policy methods**, as stated in the appendix (line 1167), while target networks are applied to all methods. (Target networks can be used with off-policy methods because training trajectories are still sampled from the current policy; the target networks only define slower-moving targets for the learnable generation process.)
> > > > 2. We highlight that **off-policy methods do not aim to estimate the KL gradient, but minimisation to 0 of an off-policy loss implies minimisation to 0 of the KL**. In particular, if the TB loss is equal to 0 under any full-support sampling distribution, then the KL divergence between the trajectory distributions associated with the generation and destruction processes is also 0. Thus **unbiased estimation of KL gradients is not the goal, but minimisation of KL is a consequence of off-policy training**. The ELBO (equal to the true log-normalising constant minus the KL), is used as a metric in our experiments and demonstrates the above empirically: **the off-policy methods have lower KL (higher ELBO) than on-policy ones that directly minimise KL**.
> > > > 3. Regarding non-stationary targets, we mitigate this using a decaying learning rate and the Adam optimiser, which together reduce the magnitude of parameter updates over time and make the targets change more slowly, thereby stabilising training in the usual way. Target networks serve the same purpose.
> > > > 4. Since training a time-discretised diffusion sampler and training a GFlowNet are equivalent tasks, relevant analysis has already been developed in [Gritsaev et al., "Optimizing backward policies...", 2025], Theorem 3.1, where convergence of the same procedure in the discrete case is proven. That analysis transfers directly to the continuous (diffusion sampling) case.
> > > >
> > > > **Empirically,** we observe the following:
> > > >
> > > > 1. Our modifications (learning the destruction process and the generation variances) improve over the baseline for all methods, both on-policy and off-policy (lines 368 and 397).
> > > > 2. Off-policy methods perform better than on-policy ones (line 377).
> > > > 4. The prioritised replay buffer is beneficial for off-policy methods, and it helps our modification more than the baseline (Table 2).
> > > > 5. For this reason, off-policy (TB-based) methods are presented as the main (“base") version of our algorithms, since they give the best performance among all considered variants.
> > > > 6. Target neural networks are also beneficial (Table 2).
> > > >
> > > > These empirical results support the claims made above.
> > > >
> > > > **Thank you again for raising these important points. We would be happy to clarify further if needed.**

---

### Official Review · Reviewer_sTsU · 2025-11-01

**Soundness:** 3
**Presentation:** 3
**Contribution:** 3
**Rating:** 6
**Confidence:** 2

**Summary:**

This paper introduces a novel framework for training diffusion samplers by jointly optimizing both the generative and destruction processes in discrete time. Unlike prior approaches that fix the destruction process or assume continuous-time SDE dynamics, the authors propose learning both forward and reverse transitions, with decoupled, state-dependent variances parameterized via neural networks. This flexibility enables better adaptation to limited-step sampling regimes and complex energy landscapes.

**Strengths:**

1. Novel joint training of generation and destruction processes in diffusion samplers, enabling improved convergence and sampling quality, especially in few-step regimes.

2. Flexible design with state-dependent, decoupled variances for both processes—only possible in discrete-time formulation—leading to enhanced adaptability to complex energy landscapes.

**Weaknesses:**

1. Limited visual results: The paper presents few qualitative or visual examples (only human faces in Fig. 4), making it difficult to fully assess sampling quality, especially in image-related tasks.

2. No discussion of limitations: The paper lacks a section acknowledging potential limitations (e.g., scalability to more complex distributions, sensitivity to architecture choices), which raises concerns about generalizability.

**Questions:**

See the weakness part.

---

> ### Author Response · Authors · 2025-11-21
>
> We thank the reviewer for their positive assessment of the paper, in particular for highlighting the novelty of jointly training the generation and destruction processes and the flexibility afforded by state-dependent, decoupled variances in discrete time.
>
> We now answer the reviewer's two concerns:
>
> > Limited visual results: The paper presents few qualitative or visual examples (only human faces in Fig. 4), making it difficult to fully assess sampling quality, especially in image-related tasks.
>
> **On image-related tasks:** We have now added additional qualitative examples for the StyleGAN3 experiment (Figures 7-11), including samples across a wider range of prompts. These new results provide a more complete visual illustration of the sampler’s behaviour and support the quantitative improvements we report (ELBO, ImageReward, and CLIP-based diversity). This substantially extends the visual evidence beyond the single figure with human faces mentioned in the review.
>
> Additionally, we provide **new quantitative results on 30 prompts**, generated automatically by a LLM to avoid cherry-picking and author bias. Full results can be seen in the table below. We stress that the prior is an unconditional model.
>
>
> |Algorithm|ELBO|Reward|CLIP diversity|
> |---|---|---|---|
> |Prior (unconditional)|-190|-1.90|0.362|
> |**(Baseline)** TB$_\\theta$ [Venkatraman et al.]|20.8|0.988|0.246|
> |**(Ours)** TB$_\\theta$ (learnt var.)|43.7|1.10|0.235|
> |**(Ours)** TB$_{\\theta,\\varphi}$|53.7|1.23|0.231|
> |**(Ours)** TB$_\\theta$|59.5|1.27|0.230|
>
> Furthermore, we find an improvement in ELBO over the baseline TB in 25 of the 30 prompts using TB(theta) with learnt variance, 29 of the 30 using TB(theta,phi), and 28 of the 30 prompts with TB(theta) + TLM(phi). **These results strengthen the conclusions in §4.4 of the paper.**
>
>
> **On synthetic tasks:** Showing differences between samplers on synthetic densities, where uninterpretable distributions in dimension >2 are modelled, is inherently difficult. However, we direct your attention to Figure 3, which illustrates well the differences between the modelled distributions on the Funnel target density. This figure allows to visually understand how learnt variance and a learnt destruction process are better than a fixed variance and destruction process.
>
> > No discussion of limitations: The paper lacks a section acknowledging potential limitations (e.g., scalability to more complex distributions, sensitivity to architecture choices), which raises concerns about generalizability.
>
> Thank you for the suggestion of adding such a section, which we have now done within the conclusion (§5). While we have already discussed limitations and scalability throughout the text, this new section combines them in one place.
>
> **Thank you again for your comments and questions. We are happy to provide any additional clarifications during the discussion period.**

---

### Official Review · Reviewer_CUPi · 2025-11-02

**Soundness:** 2
**Presentation:** 3
**Contribution:** 2
**Rating:** 2
**Confidence:** 4

**Summary:**

This paper proposes a method for training diffusion samplers, which are models designed to sample from an unnormalized density (defined by an energy function) without access to data samples. Specifically, it introduces the joint training of both the generation and destruction processes by viewing them as discrete-time policies. The core novelty is decoupling their variances, which enables the means and variances of both processes to be learned as state-dependent neural networks, rather than being fixed or constrained by continuous-time theory. The experiments demonstrate that this joint training approach results in faster convergence and improved sampling quality across various small scale benchmarks, especially when the number of generation steps is limited or the energy landscape has narrow modes. This scalability is validated on a high-dimensional GAN latent space sampling task, showing quantitative and qualitative benefits for conditional image generation.

**Strengths:**

1. The problem studies is novel and interesting
2. The proposed training objective, Second-Moment Divergence, deviates from the standard KL formulation, which is an interesting direction to explore.
3. The design space is meticulously swept over, with the key ingredients for stable training reported in this paper.
4. The advantage is most pronounced when the number of sampling steps is small. The paper shows that on some tasks, its method with as few as 5 steps can outperform 20-step samplers that use fixed variances.
5. Their experiments on synthetic dataset, though limited by their scale, are very explanatory.

**Weaknesses:**

1. Despite the non-trivial efforts to stabilize the training, the joint training process is inherently unstable. The Trajectory Likelihood Maximization (TLM) objective, one of the main candidates for generically training the destruction process, is "unstable and often leads to divergent training" when the number of steps is large.
2. The method also seems to be highly sensitive to hyperparameters. L265-266, "tuning relative learning rates is critical for stable training".
3. The results on GAN, which is regarded as scalability test of the method, has mixed results.

**Questions:**

Theoretically speaking, would the trained sampler guaranteed to approximate the target distribution as the number of sampling steps goes to infinity?

---

> ### Author Response · Authors · 2025-11-21
>
> We thank the reviewer for their thoughtful feedback. We are glad that they find the problem setting and our contribution novel and interesting and that they appreciate the thorough ablation study and design-space exploration. We are also grateful that they found our synthetic experiments clear and explanatory and that they highlight that, on some tasks, our method achieves up to a 4× reduction in inference steps compared to baselines.
>
> We proceed to answer the reviewer's questions and concerns.
>
> > Despite the non-trivial efforts to stabilize the training, the joint training process is inherently unstable. The Trajectory Likelihood Maximization (TLM) objective, one of the main candidates for generically training the destruction process, is "unstable and often leads to divergent training" when the number of steps is large.
>
> In nearly all past works in the literature on GFlowNets, from which our algorithms are inspired, where both generation and destruction policies are trained, TB training is used for both. **The version of our proposed algorithm that uses TB for both processes should be considered the "base" version, and it is stable across all experiments.**
>
> An exception is [Gritsaev et al., 2025], which studied discrete-space problems, proposing TLM as an alternative method for training the destruction process. We included TLM for completeness, with the goal of seeing whether the findings of [Gritsaev et al.] generalise to the diffusion sampling case. We find that for short trajectories TLM works well, but for long ones it is unstable.
>
> > The method also seems to be highly sensitive to hyperparameters. L265-266, "tuning relative learning rates is critical for stable training".
>
> The relative learning rate is indeed an important hyperparameter. However, in §4.2 and §4.3 all other hyperparameters are kept fixed across tasks, and we vary only the destruction learning rate. Moreover, §4.3 provides a systematic ablation of our stability-related design choices.
>
> This suggests that the method is not “highly sensitive” overall, and in practice typically requires tuning only the destruction learning rate. In addition, we observe that the optimal destruction learning rate is always less than or equal to the generative learning rate and becomes notably sensitive only for energy landscapes with a high Lipschitz constant, such as Hard Funnel.
>
> > The results on GAN, which is regarded as scalability test of the method, has mixed results.
>
> We respectfully disagree with this assessment. Our method outperforms the baseline on ELBO (104.5 vs. 98.8) and ImageReward (1.49 vs. 1.42), while matching it on CLIP diversity (0.24 vs. 0.24), as shown in Table 3. Figure 4 provides a visual comparison. **Given these results, our method is consistently stronger than the baseline.**
>
> This experiment was proposed in [Venkatraman et al., "Outsourced diffusion sampling", 2025], and we compare our proposed, stable algorithm with the baseline from [Venkatraman et al.]. We conduct experiments on seven different tasks, each defined by its own prompt, and report three metrics: ELBO, average ImageReward score, and diversity measured as the average cosine distance in CLIP space. Note that as the task is to match the posterior distribution, **ELBO is the main performance measure**: a model that always samples a single high-scoring image would achieve high reward and low diversity, while a model that does not optimise the reward, such as the prior, would achieve low reward and high diversity.
>
> To further strengthen this experiment, we provide **new results on 30 prompts**. The conclusions remain the same: our method outperforms the baseline on ELBO and ImageReward while achieving similar CLIP diversity. The new prompts were generated automatically by a LLM, not cherry-picked by us. Full results can be seen in the table below.
>
> (continued in next comment)

---

> ### Author Response · Authors · 2025-11-21
>
> (continued from previous comment)
>
> |Algorithm|ELBO|Reward|CLIP diversity|
> |---|---|---|---|
> |Prior (unconditional)|-190|-1.90|0.362|
> |**(Baseline)** TB$_\\theta$ [Venkatraman et al.]|20.8|0.988|0.246|
> |**(Ours)** TB$_\\theta$ (learnt var.)|43.7|1.10|0.235|
> |**(Ours)** TB$_{\\theta,\\varphi}$|53.7|1.23|0.231|
> |**(Ours)** TB$_\\theta$|59.5|1.27|0.230|
>
> **Furthermore, we find an improvement in ELBO over the baseline TB in 25 of the 30 prompts using TB(theta) with learnt variance, 29 of the 30 using TB(theta,phi), and 28 of the 30 prompts with TB(theta) + TLM(phi).**
>
> > Theoretically speaking, would the trained sampler guaranteed to approximate the target distribution as the number of sampling steps goes to infinity?
>
> In the case where the variances of the generation and destruction processes coincide and depend only the time variable, the continuous-time limiting behaviour was studied in [Berner et al., "From discrete-time policies...", 2025]. That paper proved that in the continuous-time limit, discrete-time training objectives, including Reverse KL (PIS) and TB, approach divergences between path space measures, implying that **integration in continuous time would sample from the target distribution when the loss is minimised in the limit**. We note that while that paper's experimental section only used fixed (pinned Brownian motion) destruction processes, the theoretical analysis did not assume any particular destruction process and is thus applicable to our case.
>
> However, **the need to learn the variances of the generation and destruction processes emerges from the choice of fixed time discretisation, and learnt variances adapt to this choice.** When it is not enforced that the variances coincide, the limit of the training objectives does not exist -- indeed, the limiting distributions are processes defined by SDEs that are not absolutely continuous with respect to one another. Thus it does not make sense to take the number of discretisation steps to infinity in this case.
>
> **Given the clarifications above and the positive aspects you emphasized -- as well as the answers above, which we believe to have clarified certain misunderstandings and addressed your concerns -- we feel that the current rating may underestimate the contribution of our work, and we respectfully ask you to consider raising the score.**

---

### Comment · Area_Chair_okSw · 2025-11-25
**Discussion for consensus**

thanks for the thoughtful reviews and for engaging in the discussion.

Given the authors’ detailed clarifications (e.g., stability vs. LR tuning, TLM vs. TB, continuous-time limits, new visual + 30-prompt results), it’d be helpful if each of you could briefly say whether these responses resolve your earlier concerns and whether your score/stance has shifted. This will help us converge on a clear consensus. Appreciate everyone’s time!

---

### Meta-Review · Area_Chair_Djah · 2026-01-02

**Summary:**

This paper aims to improve the diffusion samplers by moving away from the standard pipeline, which uses fixed destruction (forward) processes. Instead, the authors propose a method where both the generation and destruction processes are trained together as discrete-time policies. This allows the model to learn state-dependent variances for both directions through neural networks.

Although the reviewer appreciates the novel idea of "jointly training the destruction and generation process", the concerns from two reviewers, YXt4 and CUPi, seem to remain after the rebuttal. Considering the overall shortcomings raised by the reviewers, the AC recommends rejection for this version. However, the AC would like to see the authors further develop their paper and try to solve the questions raised by the reviewers.

**Reviewer Concerns:**

The initial reviews were a mix of appreciating novelty and concern over the complexity.

The reviewer CUPi has valid concerns about the stability and the sensitivity of hyperparameters. More importantly, the review pointed out that the method can not extend to continuous time scenarios, where the sampling step goes to a large number.  The rebuttal from the authors addresses partial of the concerns but also admits that the method's theoretical grounding is less "clean" than the standard diffusion process. Similarly, the reviewer YXt4 also expresses similar concern, namely, a lack of theoretical analysis regarding convergence.

Meanwhile, many reviewers acknowledged the importance and novelty of the proposed method.

**Reviewer Scores:**

Reviewer CUPi:2
Reviewer YXt4: 4
Reviewer sTsU: 6, but not informative at all.
Reviewer 83Ld: 8

After the rebuttal, the reviewers CUPi and YXt4 seem not to be convinced. Meanwhile, Reviewer 83Ld is uncertain (confidence 3) about his review and also pointed out the instability issue of the proposed method.

---

### Decision · Program_Chairs · 2026-01-26

Reject